# ELDEN: Exploration via Local Dependencies

**Zizhao Wang**[*]
University of Texas at Austin
zizhao.wang@utexas.edu

**Jiaheng Hu**[*]
University of Texas at Austin
jhu@cs.utexas.edu

**Peter Stone**[†]
University of Texas at Austin, Sony AI
pstone@cs.utexas.edu

**Roberto Martín-Martín**[†]
University of Texas at Austin
robertomm@cs.utexas.edu

## Abstract

Tasks with large state space and sparse rewards present a longstanding challenge to reinforcement learning. In these tasks, an agent needs to explore the state space efficiently until it finds a reward. To deal with this problem, the community has proposed to augment the reward function with *intrinsic reward*, a bonus signal that encourages the agent to visit *interesting states*. In this work, we propose a new way of defining interesting states for environments with factored state spaces and complex chained dependencies, where an agent's actions may change the value of one entity that, in order, may affect the value of another entity. Our insight is that, in these environments, interesting states for exploration are states where the agent is uncertain *whether* (as opposed to *how*) entities such as the agent or objects have some influence on each other. We present **ELDEN**, **E**xploration via **L**ocal **D**ep**EN**dencies, a novel intrinsic reward that encourages the discovery of new interactions between entities. ELDEN utilizes a novel scheme — the partial derivative of the learned dynamics to model the local dependencies between entities accurately and computationally efficiently. The uncertainty of the predicted dependencies is then used as an intrinsic reward to encourage exploration toward new interactions. We evaluate the performance of ELDEN on four different domains with complex dependencies, ranging from 2D grid worlds to 3D robotic tasks. In all domains, ELDEN correctly identifies local dependencies and learns successful policies, significantly outperforming previous state-of-the-art exploration methods.

## 1 Introduction

Reinforcement learning (RL) has achieved remarkable success in recent years in tasks where a well-shaped dense reward function is easy to define, such as playing video games [33, 18, 4] and controlling robots [9, 2, 15, 16]. However, for many real-world tasks, defining a dense reward function is non-trivial, yet a sparse reward function based on success or failure is directly available. For such reward functions, learning good policies is often challenging, as it requires efficient exploration of the state space.

To address this challenge, RL researchers proposed the use of an *intrinsic reward*, an additional task-agnostic signal given to the agent for visiting *interesting states*. Intrinsic reward methods can be roughly classified into two main paradigms: curiosity [20, 25, 6] and empowerment [29, 27, 14], where the agent is rewarded either for visiting novel states or for obtaining maximal control over the environment, respectively.

---

[*]Equal contribution
[†]Equal supervision

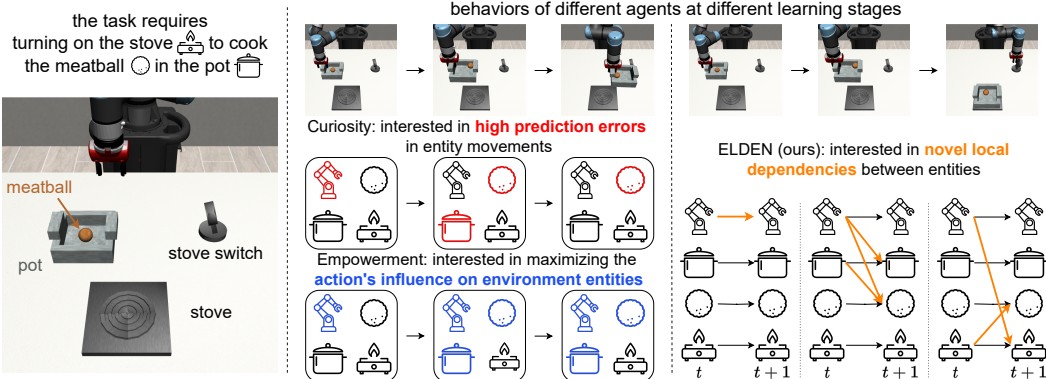

Figure 1: **(Left)** In a kitchen task with multiple potential agent-object and object-object interactions, **(Middle)** for a curiosity-based agent interested in hard-to-predict entity motion, it will initially focus on exploring arm movement, then on pot and meatball manipulation, and finally keep rolling the meatball whose outcomes are challenging to predict. On the other hand, for an empowerment-based agent interested in maximizing the action's influence, it begins with controlling the arm and then learning to move the pot and meatball simultaneously, but it ignores the potential interaction between the stove and the meatball. **(Right)** ELDEN avoids those issues by identifying whether dependencies between entities happen and focusing the exploration on novel ones. After the agent learns that it can control the pot and meatball, it will move on to explore other potential interactions, e.g., whether the stove can influence the meatball. Hence it has a larger opportunity to learn this task, compared with a curiosity or empowerment-based agent.

While these methods significantly improve exploration in some domains, there are cases where the aforementioned methods fail. Consider, for example, a kitchen environment with several objects where there are multiple potential agent-object and object-object interactions, and an agent is tasked with putting a meatball in a pot and cooking it on the stove (Fig. 1). On the one hand, curiosity-driven methods will encourage the agent to explore the environment by visiting states where the exact outcome of an action is uncertain. Consequently, for each interactable object, the agent will exhaust any possible interaction until it can accurately predict every change in the object's state. As a result, such an exploration strategy can be inefficient, especially for environments with many objects. In the kitchen example, it is hard to predict how the meatball rolls in the pot, and thus the curiosity-driven agent would keep rolling it. On the other hand, for empowerment methods, the agent is encouraged to remain in states where it can influence as many states (objects) simultaneously as possible (e.g. holding the pot with the meatball inside). By doing so, however, it ignores object-object interactions that the action cannot directly control but indirectly induce, which can be the key to task completion. In the kitchen case, an empowerment-driven agent will therefore not be interested in placing the pot and the meatball on the stove, as it forfeits control of them by doing so, even though this action enables the stove to heat the meatball. Our main insight is that, in this type of environment, an intelligently exploring agent should be able to learn that it can use the pot to move the meatball after a few trials. Then, instead of spending time learning the complex meatball movement or different styles to manipulate the pot, it would move on to explore other modes of interacting with other objects, e.g., putting the pot on the stove and switching on the stove.

Following this motivation, we propose a new definition of interesting states — focusing on *whether* the environment entities (consisting of the agents and objects) can interact, rather than *how* exactly they interact. We present ELDEN, **E**xploration via **L**ocal **D**ep**EN**dencies, a novel intrinsic reward mechanism that models the local dependencies between entities in the scene (agent-object, object-object) and uses the uncertainty about the dependencies to guide exploration. By relaxing the curiosity signal from dynamics prediction to dependencies prediction, ELDEN implicitly biases the exploration toward states where novel interaction modes happen rather than states where the state value is novel but dependencies remain the same. Specifically, ELDEN trains an ensemble of dynamics models. In each model, the local dependencies between objects are modeled by the partial derivatives of state predictions w.r.t. the current state and action. Then, the local dependency uncertainty is measured as the variance across all dynamic models.

We evaluate ELDEN on discrete and continuous domains with multiple objects leading to many interaction modes and tasks with chained dependencies. Our results show that using a partial

derivative-based extractor on dynamic models training allows us to accurately identify local connectivities between environment entities. Furthermore, the intrinsic reward derived from the identified local connectivities allows ELDEN to outperform state-of-the-art exploration methods (curiosity and empowerment-driven) on the tested domains.

## 2 Related Work

The idea of using intrinsic reward to facilitate exploration in reinforcement learning is a long-studied topic that can be dated back to Schmidhuber [25]. In this section, we first discuss two main classes of intrinsic reward related to ELDEN. Since ELDEN requires reasoning about the local dependencies between environment entities, we also discuss works that involve utilizing dependencies/causality in reinforcement learning.

### 2.1 Curiosity-Driven Exploration

Curiosity-driven exploration rewards an agent for visiting "novel" states, where different methods define the "novelness" of a state in different ways. For methods that utilize visit count to define state novelty, Bellemare et al. [3] utilized a density model to estimate pseudo-count for each state, which is then used to derive intrinsic reward for the agent; Tang et al. [30] uses locality-sensitive hashing code to convert high-dimensional states to hash codes such that the visit count for each hash code can be explicitly kept track of. For methods that utilize predictiveness to define state novelty, Stadie et al. [29] learns a forward dynamics model that operates on a learned latent space of the observation, and uses the prediction error as the intrinsic reward for exploration; Burda et al. [6] uses randomly initialized networks to extract state features, where the agent is encouraged to visit states where predictions about the state features are inaccurate; Pathak et al. [21] utilizes disagreements within an ensemble of dynamic models as a signal for intrinsic reward, and directly backpropagates from the reward signal to the policy parameters to improve exploration. Pathak et al. [20] incorporates empowerment into the curiosity-based method by learning an inverse dynamics model that maps from state to action. The inverse dynamics model defines a feature space for the states, where the prediction error of a forward dynamic model is measured in this feature space.

However, learning accurate dynamics can be difficult and require a significant coverage of a (possibly large) state-action space; e.g., when a robot manipulates a block, it would have to experience multiple action-reaction pairs to be able to make accurate predictions [13]. Furthermore, prior curiosity-driven exploration methods can be derailed by the stochasticity in the dynamics, i.e., when the outcome of an action has large entropy (e.g., tossing a coin), so the agent would keep repeating it. ELDEN can be considered a type of curiosity-driven exploration method. However, unlike previous works which only consider improving knowledge about the dynamics, ELDEN explicitly considers knowledge about local dependencies between environment entities, and utilizes it to encourage exploration. Thus, it avoids the need to learn an accurate dynamic model and is less sensitive to environmental noise and stochasticity.

### 2.2 Empowerment-Driven Exploration

Empowerment-based exploration methods are based on a different understanding of what states should be encouraged for task-agnostic exploration [35, 27, 8, 19]. Their main idea is that the most interesting states to explore for any task are states where the agent has the most controllable diversity about what the next state will be, i.e., states where there are multiple possible next states that can be chosen by the agent. From those states, it is easier to fulfill any downstream task that requires purposefully changing the state. Empowerment-based exploration methods reason about the controllable elements in the environment (states that can be influenced by agent's actions), and encourage the agent to find states where this controllability is large, typically through some form of mutual information maximization between the agent's actions and the next states. In particular, Zhao et al. [35] uses domain knowledge to divide up the state space into internal state and external state, and maximize the mutual information between them. Seitzer et al. [27] measures the local dependencies between action and environment entities by estimating their conditional mutual information (CMI) and using it as intrinsic reward signal to encourage the agent to maximize the influence of action over environment entities.

However, due to the difficulty in measuring the mutual information across a multi-step trajectory, existing empowerment-based methods only measure 1-step empowerment, e.g., how much the agent directly influences variables. Thus they cannot detect dependencies between objects (e.g., indirect tool use such as using the stove to cook meals) and the environment's influence on the agent (e.g., the elevator can take the agent to the desired floor). Furthermore, since the objective of empowerment-based methods is to maximize controllability, they tend to only control the easiest-to-manipulate object to maximize its empowerment when there are multiple controllable variables.

ELDEN is closely related to Seitzer et al. [27], but differs from it in three main aspects: first, unlike Seitzer et al. [27], which only considers the interaction between action and environment entities, ELDEN also considers the interaction between environment entities that are not locally dependant on the action, which allows curiosity about indirect interaction between the agent and the environment entities to propagate through time during the RL training. Second, unlike Seitzer et al. [27], ELDEN tries to visit states with novel interactions, instead of maximizing controllability, thus avoiding the tendency to only interact with easy-to-control objects. Lastly, ELDEN estimates local dependencies through reasoning about the partial derivatives of learned dynamic models, which we empirically show to be more accurate and computationally efficient compared to the CMI-based estimation in Seitzer et al. [27].

## 2.3 Causality in Reinforcement Learning

The concept of incorporating causality in the training of a reinforcement learning agent has been utilized in many different forms. Wang et al. [31] demonstrates that incorporating causal dependencies between environment entities can greatly improve generalization to out-of-distribution states. Hu et al. [10] exploited causal dependencies between action dimensions and reward terms to reduce variance in the gradients and facilitate policy learning of mobile manipulation tasks. Pitis et al. [24] shows that knowing the local causal dependencies between objects can facilitate policies to generalize to unseen states and actions. Pitis et al. [23] uses local dependencies to generate counterfactual samples in order to facilitate sample efficiency. Sontakke et al. [28] discovers causal factors in the dynamics of a given environment through a causal curiosity reward term.

Like in Pitis et al. [23], ELDEN learns to predict the local connectivities between environment entities depending on the state and the action values. However, we do not generate counterfactuals but create an intrinsic reward based on the local dependency that facilitates exploration with RL in sparse-reward setups.

## 3 ELDEN: Exploration via Local Dependencies

In the section, we introduce ELDEN, which infers the local dependencies between environment entities and uses the uncertainty of dependencies as an intrinsic reward for tackling hard-exploration problems. In Sec. 3.1, we formally define the problem setup of ELDEN. In Sec. 3.2, we discuss how ELDEN uncovers local dependencies. In Sec. 3.3, we describe how ELDEN improves exploration with the intrinsic reward.

### 3.1 Problem Statement

We consider decision-making as a discrete-time Markov Decision Process $(\mathcal{S}, \mathcal{A}, \mathcal{P}, \mathcal{R})$, where $\mathcal{S}$ is a state space which we assume can factored as $\mathcal{S} = \mathcal{S}^1 \times \cdots \times \mathcal{S}^N$, $\mathcal{A}$ is an action space, $\mathcal{P}$ is a Markovian transition model, and $\mathcal{R}$ is a reward function. The goal of the RL agent is to optimize the parameters $\theta$ of a policy $\pi_\theta$ such that the total expected return under $\pi_\theta$ is maximized. Specifically, we focus on cases where $\mathcal{R}$ is sparse, and therefore intelligent exploration is crucial to the discovery of optimal policies.

**Local Causal Graph Model** We can model the transition at time step $t$ as a Causal Graphical Model (CGM) [22] consisting of (1) nodes $(\mathcal{S}_t, \mathcal{A}_t, \mathcal{S}_{t+1})$, (2) a directed graph $\mathcal{G}$ describing *global* dependencies between nodes, and (3) a conditional distribution $p$ for each state variable at the next time step, $\mathcal{S}_{t+1}^n$. We assume the transition can be factorized as $\mathcal{P}(s_{t+1}|s_t, a_t) = \prod_{n=1}^N p(s_{t+1}^n|\text{Pa}(\mathcal{S}_{t+1}^n))$, where the $\text{Pa}(v)$ are parents of a node $v$ in the causal graph $\mathcal{G}$. For many environments, $\mathcal{G}$ can be dense or even fully connected, because whenever it is possible for $\mathcal{S}^j$ to depend on $\mathcal{S}^i$, no matter how

---

**Algorithm 1** Training of ELDEN (on-policy)

---

1: Initialize the dynamics ensemble $\{f\}^M$, policy $\pi_\theta$, and replay buffer $D$.
2: **for** number of training iterations **do**
3:     Collect K environment transitions $\{s_t, a_t, r_{t,\text{task}}, s_{t+1}\}^K$ with current policy $\pi_\theta$
4:     **for** $k = 1 \ldots K$ **do**
5:         $\mathcal{G}^{k,m} = f^m.\text{compute}\_\mathcal{G}(s_t^k, a_t^k)$ (Sec. 3.2)              ▷ Compute the local dependency graph
6:         $r_{t,\text{intrinsic}}^k = \text{variance}(\{\mathcal{G}^{k,m}\}_{m=1}^M)$
7:         $r_t^k = r_{t,\text{task}}^k + \beta \cdot r_{t,\text{intrinsic}}^k$
8:     **end for**
9:     Update the policy $\pi_\theta$ with $\{s_t, a_t, r_t, s_{t+1}\}^K$ (Sec. 3.3)
10:    Add $\{s_t, a_t, s_{t+1}\}^K$ into the replay buffer $D$
11:    Sample a mini-batch $B = \{s_t, a_t, s_{t+1}\}_{\text{buff}}^K$ from $D$
12:    Train the dynamics ensemble $\{f\}^M$ with $B$ (Sec. 3.2)
13: **end for**

---

unlikely, it is necessary to include the edge $\mathcal{S}^i \to \mathcal{S}^j$. However, in the real world, even if possible to interact, most entities are independent of each other most of the time. Following this observation, we are interested in inferring *local* dependencies that are specific to $(s_t, a_t)$, represented by a local causal graph $\mathcal{G}_t$ that is minimal by removing inactive edges in $\mathcal{G}$.

## 3.2 Identifying Local Dependencies with Dynamics Partial Derivatives

Based on the definition in Sec. 3.1, the key component of ELDEN is to accurately evaluate which potential dependencies between environment entities are locally active, i.e., identify the local causal graph $\mathcal{G}_t$ given $(s_t, a_t)$. This identification requires answering a challenging question — whether entity $i$'s value, $\mathcal{S}_t^i = s_t^i$ is the cause of entity $j$ to have value $\mathcal{S}_{t+1}^j = s_{t+1}^j$. ELDEN approaches it with the inspiration from the *but-for* test: the local dependency exists if $\mathcal{S}_{t+1}^j = s_{t+1}^j$ would not happen but for $\mathcal{S}_t^i = s_t^i$. In other words, we assess whether $\mathcal{S}_{t+1}^j$ would change if $\mathcal{S}_t^i$ has a different value. Notice that, since we focus on *local* dependencies, we only want to vary $\mathcal{S}_t^i$ near its actual value $s_t^i$ rather than trying all its possible values.

To this end, ELDEN utilizes partial derivatives to identify local dependencies, as they naturally capture the extent of change in $\mathcal{S}_{t+1}^j$ with respect to $\mathcal{S}_t^i$. Specifically, assuming the access of ground truth transition probability $p$ (which we will relax later), ELDEN considers $\mathcal{S}_{t+1}^j = s_{t+1}^j$ locally depends on $\mathcal{S}_t^i = s_t^i$ if

$$\left| \frac{\partial p(s_{t+1}^j | s_t, a_t)}{\partial s_t^i} \right| \geq \epsilon, \tag{1}$$

where $\epsilon$ is a predefined threshold. A large partial derivative indicates that a slight change in $\mathcal{S}_t^i$ will lead to a substantial change in $\mathcal{S}_{t+1}^j$, thus satisfying the but-for test.

To evaluate partial derivatives without the ground truth transition probability, ELDEN approximates $p$ with a dynamics model $f$ parameterized by a neural network $\hat{p}(s_{t+1}^j) = f(s_t, a_t)$ and trains $f$ by maximizing the log-likelihood of $\hat{p}(s_{t+1}^j)$ (for notational simplicity, we omit the conditionals $s_t, a_t$ in $p$ in this section). Due to limited data and training errors, even when two environment entities are not locally dependent, there occasionally exists a large partial derivative between them. To reduce such false positives, ELDEN further applies regularization to suppress partial derivatives w.r.t. inputs that are not necessary for predicting $s_{t+1}^j$. The overall loss of dynamics training is

$$L_f = -\log \hat{p}(s_{t+1}^j) + \lambda \sum_{i,j} \left| \frac{\partial \hat{p}(s_{t+1}^j)}{\partial s_t^i} \right|, \tag{2}$$

where $\lambda$ is the regularization coefficient.

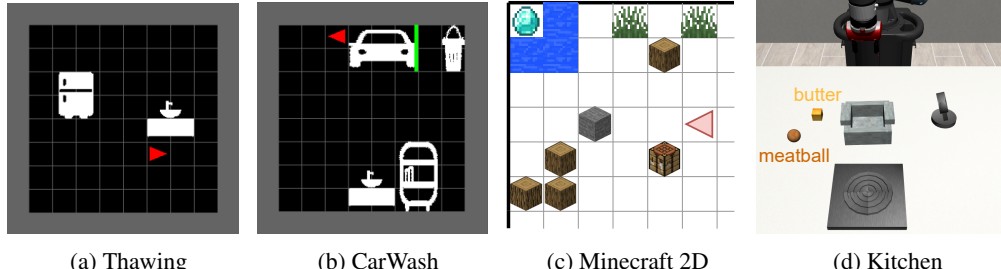

| (a) Thawing | (b) CarWash | (c) Minecraft 2D | (d) Kitchen |

Figure 2: We test ELDEN on three domains and four environments. **(a) (b)** Mini-behavior and **(c)** Minecraft 2D with discrete state spaces, where the agent has to achieve a series of temporally extended tasks with complex object interactions. **(d)** Robosuite, a robot table-top manipulation simulation environment with continuous state spaces, where the robot needs to perform multiple interdependent subtasks to finish the cooking task.

### 3.3 ELDEN Policy Learning

ELDEN utilizes the local dependency identification described in Sec. 3.2 to improve exploration for model-free RL. The key idea behind ELDEN is to encourage an agent to visit states where new local dependencies are likely to emerge. When the ground truth local dependencies are available, the novelty of local dependencies between state variables can be measured by the magnitude of error of the dependencies identified by our method. Unfortunately, in many cases, it is hard to manually specify the ground truth local dependencies. Instead, ELDEN trains an ensemble of dynamics models, and measures dependency novelty by the variance of the local dependency graphs extracted independently from each of the dynamics models in the ensemble. Specifically, ELDEN first computes the variance of each edge in the graph and then uses the mean of the edge variance as the graph variance. Finally, the calculated variance is used as the intrinsic reward and is scaled with a coefficient $\beta$ that controls the magnitude of the exploration bonus, and added to the task reward $r_{\text{task}}$.

ELDEN is applicable to both on-policy and off-policy settings. We show the pseudo-code for on-policy ELDEN in Algorithm 1. The off-policy version of ELDEN can be easily derived by updating the policy with transitions sampled from the replay buffer in line 9 of Algorithm 1. Importantly, any model-free RL algorithm can be used for the policy update step.

## 4 Experiments

In our experiments, we aim to answer two main questions: *Q1:* Is ELDEN able to accurately detect local dependencies between environment entities in factored state spaces (Sec. 4.1)? *Q2:* Does ELDEN improve the performance of RL algorithms in sparse-reward environments with chained dependencies (Sec. 4.2)?

**Environments** As shown in Fig. 2, we evaluate ELDEN in four simulated environments with different objects that have complex and chained dependencies: (1) CARWASH, (2) THAWING, (4) 2D MINECRAFT and (3) KITCHEN. Both CARWASH and THAWING are long-horizon household tasks in discrete gridworld from the Mini-BEHAVIOR Benchmark [12]. MINECRAFT 2D is an environment modified from the one used by Andreas et al. [1], where the agent needs to master a complex technology tree to finish the task. KITCHEN is a continuous robot table-top manipulation domain implemented in RoboSuite [36]. To complete tasks in these environments and receive the sparse reward, the agent has to conduct a series of actions that change not only the state of the interacted entities but also induce further interaction between interacted entities and others (e.g., interacting with the stove switch that enact interaction between the stove and the cooking the meatball). The agents in all environments can select between a set of action primitives, a set of discrete actions that can be applied on each object, e.g., `goTo(obj)` or `pick(obj)`. Notice that even with the action primitives, these domains are still very hard to solve due to the presence of many interaction modes and the difficulty in finding the correct (potentially long) sequence of interactions among many options that will lead to task success. We provide further descriptions of each environment in Appendix Sec B.

Table 1: Mean $\pm$ std. error of ROC AUC ($\uparrow$) and F1 ($\uparrow$) of local dependency prediction

| | THAWING | | CARWASH | | KITCHEN | |
|---|---|---|---|---|---|---|
| | ROC AUC | F1 | ROC AUC | F1 | ROC AUC | F1 |
| ELDEN | **0.71** $\pm$ 0.01 | 0.57 $\pm$ 0.00 | **0.78** $\pm$ 0.02 | 0.66 $\pm$ 0.02 | **0.66** $\pm$ 0.01 | 0.25 $\pm$ 0.01 |
| pCMI | 0.55 $\pm$ 0.01 | 0.60 $\pm$ 0.00 | 0.73 $\pm$ 0.02 | **0.78** $\pm$ 0.01 | 0.60 $\pm$ 0.00 | **0.28** $\pm$ 0.00 |
| Attn | 0.65 $\pm$ 0.04 | **0.63** $\pm$ 0.01 | 0.66 $\pm$ 0.01 | 0.55 $\pm$ 0.03 | 0.51 $\pm$ 0.01 | 0.22 $\pm$ 0.02 |
| Input Mask | 0.50 $\pm$ 0.00 | 0.40 $\pm$ 0.00 | 0.50 $\pm$ 0.00 | 0.32 $\pm$ 0.01 | 0.50 $\pm$ 0.00 | 0.08 $\pm$ 0.00 |
| Attn Mask | 0.45 $\pm$ 0.03 | 0.47 $\pm$ 0.02 | 0.47 $\pm$ 0.07 | 0.43 $\pm$ 0.03 | 0.52 $\pm$ 0.01 | 0.13 $\pm$ 0.01 |

**Implementation Details** For discrete state or action spaces, the partial derivatives w.r.t. $s_t/a_t$ are undefined. To address this issue, we use Mixup [34] to create synthetic inputs and labels by linearly combining pairs of inputs and labels, thus approximately changing the input space to be continuous. Compared to learning from discrete inputs only, dynamics models trained on such data generate partial derivatives that better reflect local dependencies, as shown in Sec 4.3.1. For the 2D MINECRAFT and KITCHEN environments where some local dependencies have complex preconditions and thus are hard to induce, we apply sample prioritization to dynamics learning, where the priority is measured as prediction error. In this way, the dynamics model gets aware of unknown interactions faster and guides the exploration more efficiently than not using prioritization. Further details are provided in the Appendix.

## 4.1 Evaluating the Detection of Local Dependencies

We compare the local dependencies extracted by ELDEN with the following baselines (see implementation details in Appendix Sec. C):

- **pCMI** (point-wise conditional mutual information) [27, 31]: defined as $\log \frac{p(s_{t+1}^j | s_t, a_t)}{p(s_{t+1}^j | s_t \setminus s_t^i, a_t)}$. It quantifies how likely it is that $s_{t+1}^j$ depends on $s_t^i$.

- **Attn** (attention): Use the score between each entity pair computed by the attention modules inside the dynamics model to quantify local dependencies.

- **Input Mask**: we implement a learnable binary mask to the dynamics model that can zero out some inputs conditioned on $(s_t, a_t)$: $f([s_t, a_t] \odot M(s_t, a_t))$. During training, the mask is regularized to use as few inputs as possible with L1 regularization, leading to a quantification of minimal local dependencies.

- **Attn Mask**: we implement a learnable mask to the dynamics model similar to the one in Input Mask, but in this case, the mask is applied to the attention scores. The mask is regularized following the method by Weiss et al. [32].

We train the dynamics model of each method with three random seeds on pre-collected transition data and evaluate their performance by predicting the local causal graph $\mathcal{G}_t$ for 50 unseen episodes based on the state-action pair $(s_t, a_t)$. We compare their predictions with the ground truth local dependencies extracted from the simulator. In the three environments, many potential local dependencies are inactive most of the time, and thus only a small portion ($\leq 3\%$) of the ground truth labels indicate the existence of local dependencies for a given entity pair. To account for such imbalance, we use the area under the receiver operating characteristic curve (ROC-AUC) and the best achievable F-score (F1) as evaluation metrics.

The results of the evaluation on the detection of local dependencies are summarized in Table 1. ELDEN outperforms all baselines in terms of ROC-AUC consistently across all environments (Q1). For the F1 score, pCMI performs best in most environments (especially in the more complex CARWASH and KITCHEN), but ELDEN performs comparably or achieves the second-best F1 scores with much less computation: pCMI computation cost is $N$ times higher than ELDEN, where $N$ is the number of environment entities, and thus pCMI scales badly to environments with a large number of objects. Further evaluation details can be found in the Appendix.

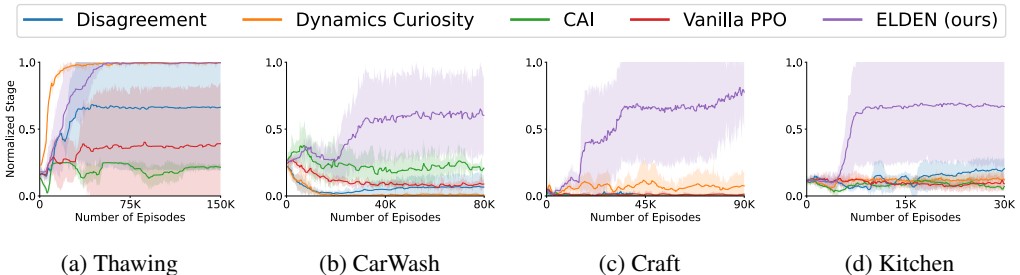

(a) Thawing  (b) CarWash  (c) Craft  (d) Kitchen

Figure 3: Learning curve of ELDEN (ours) compared to baseline approaches. Each method uses three random seeds, and we show the mean ± std dev of the number of stages completed toward task success. The stage count is normalized to [0, 1], where 1 corresponds to task completion. ELDEN learns successful policies in all four test environments, and is the only method that succeeds in the *CarWash*, *2D Minecraft*, and *Kitchen* environments with complex chained dependencies.

## 4.2 Evaluating Exploration in Sparse-Reward RL Tasks

The ultimate goal of our method is to improve exploration for RL in sparse-reward setups. In the second evaluation, we compare the performance of ELDEN against several state-of-the-art intrinsic-motivation exploration algorithms in reinforcement learning, including:

- **Disagreement** [21]: the intrinsic reward is computed based on the variance of the predictions from an ensemble of forward dynamics models.

- **Dynamics Curiosity** [5]: intrinsic reward is computed based on the prediction error of a trained forward dynamics model.

- **CAI** (Causal Influence Detection) [27]: an empowerment-based method, where the agent is given intrinsic reward for maximizing the number of state entities that depend on its action.

- **Vanilla PPO** [26]: baseline without intrinsic reward that serves as control signal.

While ELDEN can be used with any RL algorithm, in our experiments we use proximal policy optimization (PPO) [26], as well as with the baselines. To facilitate the introspection of the results, we define manually a set of semantic stages representing internal progress toward task completion. The stage definitions for each of the environments are described in detail in the Appendix. Notice that these stages are not used by the agents during training and do not provide any additional reward; they are only used to facilitate the analysis of the results.

Fig. 3 depicts the count of reached stages per episode during training for each task, normalized by the number of stages to complete the task. In the normalized stage count, a value of 1 corresponds to successfully completing the task and it is the only stage where the learning agents receive sparse task reward (not intrinsic). Fig. 3 indicates that ELDEN is able to learn successful policies in all four environments. Importantly, in CARWASH, 2D MINECRAFT and KITCHEN, ELDEN is the only method that successfully learns to complete the task, demonstrating the advantage of ELDEN over the baseline algorithms in tackling tasks with complex chained dependencies (Q2).

The normalized stage count of ELDEN in CARWASH, 2D MINECRAFT and KITCHEN does not converge to 1 (completing the entire task in all episodes) mainly due to two reasons: First, in both tasks, the locations of the objects are randomly initialized at the start of each episode. For some initialization (e.g. a target object is blocked by unmovable obstacles), the task is impossible to solve. Second, in both tasks, two out of the three ELDEN training procedures with different random seeds converge to succeeding most of the time, but the training process with one seed fails to find a good policy, dragging down the mean value of the normalized stage count. This large variance in success is a current limitation of ELDEN.

In the relatively simple THAWING environment, we found ELDEN does not provide a significant advantage over the other baseline methods. The **Dynamics Curiosity** baseline learns faster to achieve the task indicating a better sample efficiency. This was rather expected: as with any exploration heuristic, ELDEN is not universally better than previous intrinsic reward methods — instead, it is better suited for a specific type of environment, where there are many complex and chained object

Table 2: Ablation of ELDEN on local dependency prediction (mean $\pm$ std. error of ROC AUC and F1)

| | THAWING | | CARWASH | | KITCHEN | |
|---|---|---|---|---|---|---|
| | ROC AUC ($\uparrow$) | F1 ($\uparrow$) | ROC AUC ($\uparrow$) | F1 ($\uparrow$) | ROC AUC ($\uparrow$) | F1 ($\uparrow$) |
| no Mixup & no Reg | $0.48 \pm 0.01$ | $0.42 \pm 0.01$ | $0.44 \pm 0.00$ | $0.27 \pm 0.01$ | N/A | N/A |
| no Reg, i.e., $\lambda = 0$ | $0.57 \pm 0.01$ | $0.52 \pm 0.01$ | $0.54 \pm 0.01$ | $0.42 \pm 0.01$ | $0.64 \pm 0.01$ | $0.24 \pm 0.01$ |
| $\lambda = 10^{-1}$ | $0.68 \pm 0.00$ | $\mathbf{0.57} \pm 0.00$ | $0.73 \pm 0.01$ | $0.58 \pm 0.02$ | $0.55 \pm 0.00$ | $0.14 \pm 0.00$ |
| $\lambda = 10^{-2}$ | $\mathbf{0.71} \pm 0.01$ | $\mathbf{0.57} \pm 0.00$ | $0.76 \pm 0.01$ | $0.60 \pm 0.00$ | $0.60 \pm 0.01$ | $0.21 \pm 0.01$ |
| $\lambda = 10^{-3}$ | $0.64 \pm 0.01$ | $0.55 \pm 0.01$ | $\mathbf{0.78} \pm 0.02$ | $\mathbf{0.66} \pm 0.02$ | $0.65 \pm 0.00$ | $0.24 \pm 0.01$ |
| $\lambda = 10^{-4}$ | $0.65 \pm 0.02$ | $0.55 \pm 0.01$ | $0.75 \pm 0.01$ | $0.60 \pm 0.01$ | $\mathbf{0.66} \pm 0.00$ | $\mathbf{0.25} \pm 0.01$ |
| $\lambda = 10^{-5}$ | $0.63 \pm 0.00$ | $0.53 \pm 0.01$ | $0.72 \pm 0.00$ | $0.57 \pm 0.00$ | $0.65 \pm 0.01$ | $0.24 \pm 0.01$ |

dependencies, not the case for THAWING. We provide additional experimental evaluations of the failure cases of ELDEN in Appendix Sec E.

### 4.3 Ablation Studies

We ablate different components of ELDEN to examine their importance to the overall methods.

#### 4.3.1 Ablations for Local Dependency Detection

In our ablation study on ELDEN for local dependency detection, we investigate the impact of each component with the following variations:

- No Mixup & No Reg: We disable the use of Mixup for discrete space prediction, and no partial derivative regularization is applied in this case.

- Different partial derivative regularization coefficients: we test with different $\lambda$ values in $\{0, 10^{-1}, 10^{-2}, 10^{-3}, 10^{-4}, 10^{-5}\}$.

As shown in Table. 2, in Thawing and CarWash environments, partial derivative regularization with appropriate coefficients significantly improves ELDEN's detection of local dependencies, compared to no regularization (i.e., $\lambda = 0$) or inappropriate $\lambda$ values. Furthermore, in discrete-state environments, Mixup smooths the landscape of partial derivatives by providing synthesized continuous inputs as exemplified in Fig. 1(b) of Zhang et al. [34], thus facilitating local dependency prediction — even when compared to using Mixup without any regularization, not using Mixup leads to a noticeable degradation in the prediction performance.

#### 4.3.2 Ablations for Task Learning

Next, we examine how different components and hyperparameters of ELDEN affect task learning:

**Ablation of Local Dependency Metrics** We compare the exploration performance when using different local dependency detection methods. Specifically, we compare with pCMI as it achieves the best local dependency detection in Sec. 4.1. We present the comparison results between ELDEN and pCMI in the Kitchen environment in Fig. 4(a) where both methods successfully learn to solve the task. However, it is important to notice that the computation cost of pCMI is $N$ times more than that of ELDEN (where $N$ is the number of environment entities), and thus may not scale to environments with a large number of entities.

**Ablation of Dynamics Sample Prioritization** We study the effectiveness of applying sample prioritization in dynamics model training. Specifically, we test ELDEN with and without prioritization in the Kitchen environment, and show the result in Fig. 4(b). We can see that ELDEN without prioritization fails to learn a useful policy. The reason is that some key entity interactions occur rather rarely before the agent masters them, e.g., frying meatball with butter. In such cases, the dynamics model needs to quickly learn that unknown dependencies appear so that it can bias the exploration toward reproducing such dependencies. Sample prioritization helps the dynamics model learn such

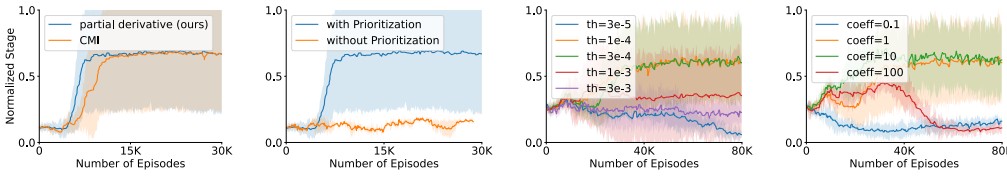

| (a) Dependency detection | (b) Sample prioritization | (c) Gradient thresholds | (d) Reward coefficients |

Figure 4: Ablation of ELDEN on task learning. Each curve uses three random seeds and shows the mean $\pm$ std dev of the normalized stages. We found ELDEN to have moderate tolerance towards hyperparameters. We found sample prioritization in dynamics learning to be crucial to the performance of ELDEN.

infrequent dependencies quickly, making it critical in environments with novel and hard-to-induce local dependencies.

**Ablation of Partial Derivative Threshold:** The partial derivative threshold $\epsilon$ determines the dependency predictions. A threshold that is too large / too small will make all dependency predictions negative / positive respectively, leading to deteriorated performance. In this section, we examine whether our method is sensitive to the choice of threshold in the CarWash environment, where the results are presented in Fig. 4(c). We observe that our method is relatively sensitive to the choice of threshold, and an inappropriate threshold could cause catastrophic failure. A potential next step for ELDEN is to automatically determine the partial derivative threshold.

**Ablation of Intrinsic Reward Coefficient:** The intrinsic reward coefficient controls the scale of the intrinsic reward relative to the task reward. We examine the effect of this coefficient by experimenting with different values in the CarWash environment, where the results are presented in Fig. 4(d). We find that our methods work well in a large range of the intrinsic reward coefficients (1 - 10), since the task only gives sparse rewards and the intrinsic rewards are the only learning signal most of the time. The only exceptions are (1) when the intrinsic reward coefficient is too large (e.g., 100), the intrinsic reward significantly surpasses the task reward, and (2) when the coefficient is too small (e.g., 0.1), the episode intrinsic reward is too small (e.g., 0.03) for PPO to learn any useful policy.

## 5    Limitations and Conclusion

We introduce ELDEN, a method for improving exploration in sparse reward reinforcement learning tasks. ELDEN identifies local dependencies between environment entities and uses the uncertainty about such dependencies as an intrinsic reward to improve exploration. Experiments demonstrate that ELDEN uncovers local dependencies more accurately compared to related methods, and significantly outperforms previous exploration methods in tasks with complex chained dependencies.

However, ELDEN is not without limitations. First, ELDEN intentionally bias exploration towards "covering up the possible interactions between objects" rather than "becoming an expert at manipulating a particular object". While such an inductive bias works well in many practical domains, it may fail when facing tasks that require precise object interaction (e.g. rotating the meatball in the pot to a specific orientation). A future direction to alleviate this problem and expand the scope of solvable tasks is to combine ELDEN with dynamics curiosity and formulate a composite intrinsic reward. Second, as noted in the experiment section, the variance of ELDEN across different random seeds can be large, While the high variance is a general problem to Reinforcement Learning, finding ways to further stabilize ELDEN can be an important direction for future work.

**Acknowledgements**   This work has taken place in the Robot Interactive Intelligence Lab (RobIn) and Learning Agents Research Group (LARG) at the Artificial Intelligence Laboratory, The University of Texas at Austin. LARG research is supported in part by the National Science Foundation (FAIN-2019844, NRT-2125858), the Office of Naval Research (N00014-18-2243), Army Research Office (E2061621), Bosch, Lockheed Martin. Both LARG and RobIn are supported by Good Systems, a research grand challenge at the University of Texas at Austin. The views and conclusions contained in this document are those of the authors alone. Peter Stone serves as the Executive Director of Sony AI America and receives financial compensation for this work. The terms of this arrangement have been reviewed and approved by the University of Texas at Austin in accordance with its policy on objectivity in research. We thank Bo Liu and Caleb Chuck for their valuable feedback on the manuscript.

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

# A    ELDEN details

**Assumptions**    We summarize our assumptions on the MDP as follows:

1. The state space can be factored as $\mathcal{S} = \mathcal{S}^1 \times \cdots \times \mathcal{S}^N$.
2. The transition of each state factor is independent, i.e., the dynamics can be represented $\mathcal{P}(s_{t+1}|s_t, a_t) = \prod_{n=1}^{N} p(s_{t+1}^n | \mathrm{Pa}(\mathcal{S}_{t+1}^n))$.
3. There is no instantaneous dependency between state factors at the same time step $t$, i.e., no dependency such as $s_t^i \rightarrow s_t^j$ for any $i, j$.

For assumption 1, factored state space is commonly employed in causality literature and applies to many simulated or robotics environments. In cases where low-level observations or partial observability are present, disentangled representation or causal representation methods can be utilized to learn a factored state space [17]. When a factored state space is available, assumptions 2 and 3 generally hold.

**Network Architecture**    In Figure 5(a), the architecture of ELDEN for predicting each state factor $s_{t+1}^j$ is illustrated. The process consists of the following steps:

1. Feature Extraction: For each input state factor $s_t^i$, ELDEN utilizes a separate multi-layer perception (MLP) to extract its corresponding feature $g^i$.
2. Entity Interaction: ELDEN employs a multi-head self-attention module to model entity interactions and generates a set of transformed features $h^i$ that incorporate information from other state factors.
3. Prediction using Multi-Head Attention: With $h^j$ as the query, ELDEN utilizes a multi-head attention module to compute the prediction $\hat{p}(s_{t+1}^j|s_t, a_t)$ for each state factor. For continuous state factor, $\hat{p}(s_{t+1}^j)$ is modeled as a normal distribution with the mean computed by the network and a fixed variance equal to 1. For discrete factor, $\hat{p}(s_{t+1}^j)$ is a categorical distribution with network outputs as class probabilities.

Throughout the prediction process, there are a total of $N$ such networks in ELDEN, with each network responsible for predicting a separate state factor $s_{t+1}^j$.

The training loss for the dynamics model is:

$$L = -\log \prod_{j=1}^{N} \hat{p}(s_{t+1}^j|s_t, a_t) + \lambda \sum_{i,j} \left| \frac{\partial \hat{p}(s_{t+1}^j)}{\partial s_t^i} \right|, \tag{3}$$

where $\lambda$ is the coefficient for partial derivative regularization.

# B    Environment Details

In this section, we provide a detailed description of the environment, including its semantic stages representing internal progress toward task completion, state space, and action space. We also highlight that while each task consists of multiple semantic stages, agents do not have access to this information. The learning signal for agents is solely based on a sparse reward of 0 or 1, indicating whether the task has been completed or not. Additionally, in each environment, the poses of all environment entities are randomly initialized for each episode.

Meanwhile, as ELDEN focuses on exploring novel local dependencies between environment entities, in all environments, the action space consists of hard-coded skills to increase the probability of entity interactions and bypass navigation challenges under sparse rewards. Extending ELDEN to explore local dependency and learn such skills simultaneously would be an important direction for future work.

**Thawing**    As shown in Fig. 6(a), the Thawing environment consists of a sink, a refrigerator, and a frozen fish. The task requires the agent to complete the following **stages**: (1) open the refrigerator, (2) take the frozen fish out of the refrigerator, and (3) put the fish into the sink to thaw it. The discrete state space consists of (i) the agent's position and direction, (ii) the positions of all environment entities,

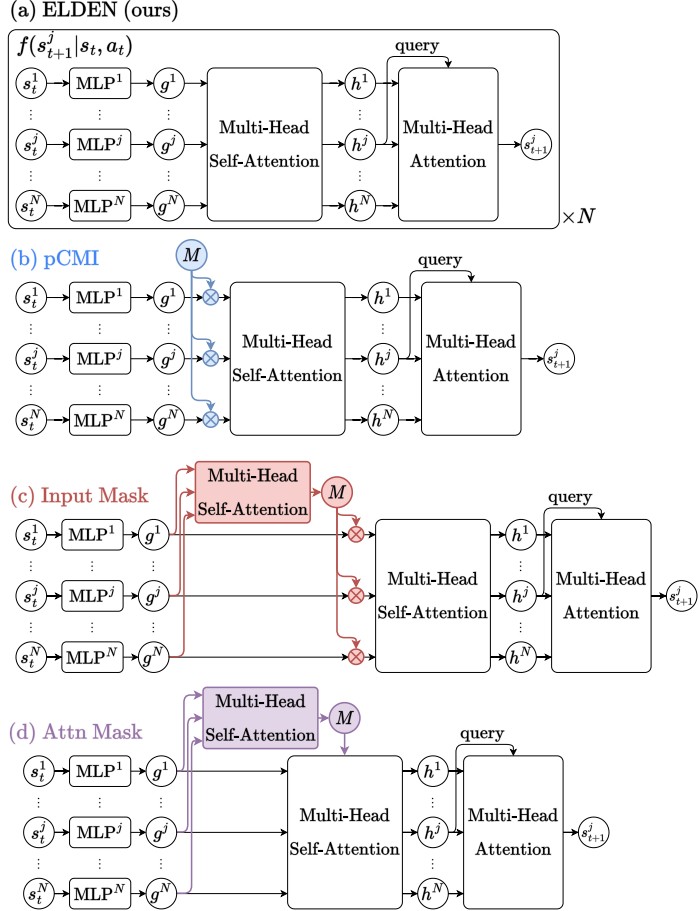

Figure 5: The dynamics model of each local dependency detection method. **(a)** The dynamics model of ELDEN for predicting $s_{t+1}^j$. Notice that each network predicts $s_{t+1}^j$ only, and there are $N$ such networks in total, each responsible for predicting one state factor in $s_{t+1}$. For visual simplicity, the "$\times N$" symbol is only shown in (a). **(b)** pCMI computes $p(s_{t+1}^j | s_t, a_t)$ and $p(s_{t+1}^j | s_t \setminus s_t^i, a_t)$ by manually setting the binary mask $M$ to different values, where $\otimes$ represents element-wise multiplication. **(c)** For Input Mask, $M$ is learned to condition on $(s_t, a_t)$ and is regularized to use as few inputs as possible. **(d)** For Attn Mask, $M$ also conditions on $(s_t, a_t)$ but is applied to the attention score in the self-attention module.

(iii) the thawing status of the fish, and (iv) whether the refrigerator door is opened. The discrete action space consists of (i) moving to a specified environment entity, (ii) picking up / dropping down the fish, and (iii) opening / closing the refrigerator door.

**CarWash**   As shown in Fig. 6(b), the CarWash environment consists of a car, a sink, a bucket, a shelf, a rag, and a piece of soap. The task requires the agent to complete the following **stages**: (1) take the rag off the shelf, (2) put it in the sink, (3) toggle the sink to soak the rag up, (4) clean the car with the soaked rag, (5) take the soap off the self, and (6) clean the rag with the soap inside the bucket. The discrete state space consists of (i) the agent's position and direction, (ii) the positions of all environment entities, (iii) the soak status of the rag, (iv) the cleanness of the rag and the car, and (iv) whether the sink is toggled. The discrete action space consists of (i) moving to a specified environment entity, (ii) picking up / dropping down the rag, (iii) toggling the sink, and (iii) picking up / dropping down the soap.

**2D Minecraft**   As shown in Fig. 6(c), the environment has complex chained dependencies — to get the gem, the agent needs to

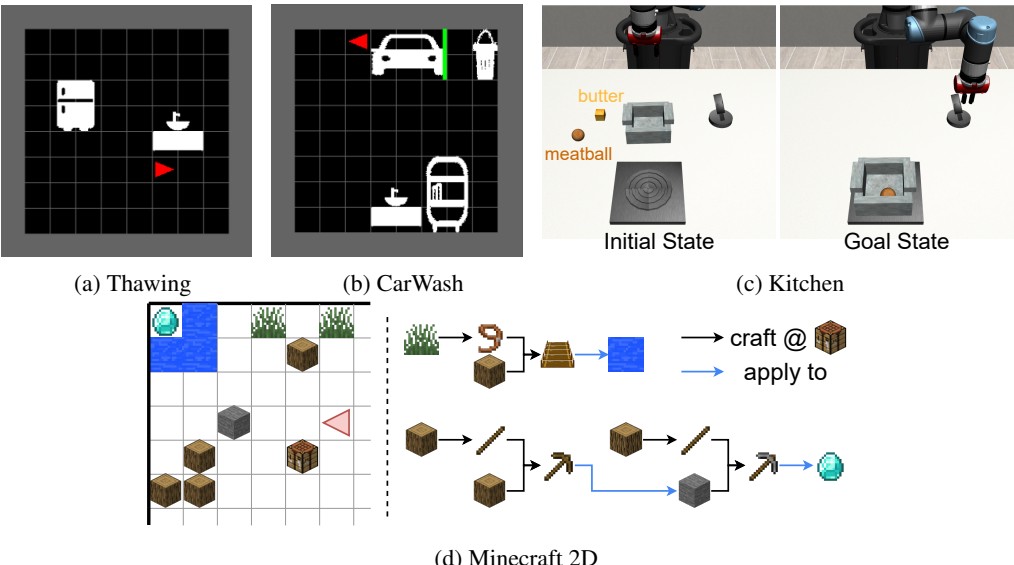

(a) Thawing  (b) CarWash  (c) Kitchen

(d) Minecraft 2D

Figure 6: Environments.

1. get across the river to reach the gem by
   (a) collecting a unit of grass and crafting a rope,
   (b) collecting a unit of wood and crafting a bridge with the rope,
   (c) building the bridge on top of the river;
2. collect the gem by
   (a) collecting a unit of wood to craft a wood stick
   (b) collecting another unit of wood and combining it with the stick to craft a wood pickaxe that is required for collecting the stone,
   (c) collecting a unit of wood and a unit of stone to craft a stick and then a stone pickaxe that is required for collecting the gem,
   (d) collecting the gem with the stone pickaxe.

Notice that all crafting must be conducted at the crafting table. The discrete state space consists of (i) the agent's position and direction, (ii) an inventory tracking the number of materials and tools that the agent has, and (iii) the positions of all environment entities. The discrete action consists of (i) picking up / applying tools to (only effective when the agent faces an environment entity and has the necessary tools to interact with it), (ii) crafting a specified tool (only effective when the agent has enough materials and faces the crafting table), and (iii) moving to a specified environment entity.

**Kitchen**  As shown in Fig. 6(d), in the kitchen environment, there are a robot arm (i.e., the agent), a piece of butter, a meatball, a pot, and a stove with its switch. The task requires the agent to complete the following **stages**: (1) pick and place the butter into the pot, (2) pick and place the pot onto the stove, (3) turn on the stove to melt the butter in the pot, (4) pick and place the meatball into the pot to cook it, and (5) turn off the stove. Notice that melting the butter is a prerequisite for cooking the meatball, otherwise, it will result in the meatball being overcooked and the task failing. The state space is continuous, consisting of the pose of all objects, the melting status of the butter, and the cooking status of the meatball (whether it is raw, cooked, or overcooked). The action space is discrete, consisting of hard-coded skills: moving to [butter, meatball, pot, pot handle, stove, stove switch], grasping, dropping, and toggling the switch. Grasping and toggling are only applicable when the end-effector is close to the corresponding environment entities.

## C  Implementation of Local Dependency Detection

**Baselines**  We give a detailed description of each baseline as follows:

- **pCMI** (point-wise conditional mutual information): it considers that the local dependency $s_t^i \to s_{t+1}^j$ exists if their point-wise conditional mutual information is greater than a predefined threshold, i.e., $\text{pCMI}^{i,j} := \log \frac{p(s_{t+1}^j | s_t, a_t)}{p(s_{t+1}^j | s_t \setminus s_t^i, a_t)} \geq \epsilon$. As shown in Fig. 5(b), to compute $\text{pCMI}^{i,j}$, Wang et al. [31] uses a manually defined binary mask $M \in [0,1]^N$ to ignore some inputs when predicting $s_{t+1}^j$: (1) to compute $p(s_{t+1}^j | s_t, a_t)$, $M$ uses all inputs (all its entries are set to 1), and (2) to compute $p(s_{t+1}^j | s_t \setminus s_t^i, a_t)$, the entry for $g^i$ is set to 0. When evaluating the local dependency, pCMI needs to compute $p(s_{t+1}^j | s_t \setminus s_t^i, a_t)$ for every $i$, and thus its computation cost is $N$ times larger than ELDEN. We also computes pCMI following Seitzer et al. [27], which yields similar performance but is even more computationally expensive compared to the method proposed by Wang et al. [31].

- **Attn** (attention): it uses the same architecture as ELDEN that is shown in Fig. 5(a). When computing the overall attention score, it averages the attention score across all heads in each module, then computes the likelihood of dependency $s_t^i \to s_{t+1}^j$ as $\sum_{k=1}^N c^{g^i, h^k} \cdot c^{h^k, s_{t+1}^j}$ where $c^{a,b}$ is the averaged score between the input $a$ and the output $b$.

- **Input Mask**: as shown in Fig. 5(c), it also uses a binary mask $M$ except that $M$ is computed from $(s_t, a_t)$. During training, to only use necessary inputs for $s_{t+1}^j$ prediction, $M$ is regularized with the L1 norm on its number of non-zero entries. The Gumbel reparameterization is used to compute the gradient for the binary $M$ [11].

- **Attn Mask**: as shown in Fig. 5(d), similar to Input Mask, a mask $M$ of size $N \times N$ is computed from $(s_t, a_t)$, but it is applied to the attention score. The mask is regularized with Stochastic Kernel Modulated Dot-Product (SKMDP) proposed by Weiss et al. [32].

For modules that are shared by all methods, we use the same architecture for a fair comparison.

**Data** For a fair comparison, when training each method, we use the same dataset collected by a scripted policy, rather than let each method collect its own data, to avoid potential performance differences caused by data discrepancies. Specifically, we use a scripted policy to expose all potential local dependencies and collect 500K transitions in each environment.

Notice that, in exploration with sparse reward experiments, the dynamics models are still trained online, using the transition data collected on its own.

**Hyperparameters** The hyperparameters used for evaluating local dependency detection of each method are provided in Table 3. Unless specified otherwise, the parameters are shared across all environments.

# D   Evaluating Exploration in Sparse-Reward RL Tasks

## D.1   Implementation

During policy learning, all methods share the same PPO and training hyperparameters, provided in Table 4. The hyperparameters for dynamics model setup during policy learning are the same as in Table 3 unless specified otherwise.

## D.2   Success Rate Plots

As a supplementary to the normalized stage metric used in the main paper, we provide the success rate as an additional metric. The success rate learning curves of all methods in the three environments are shown in in Fig. 7. Again, ELDEN outperforms and performs comparably with all baselines. Notice that, in the CarWash and Kitchen environments, all baselines never succeed throughout the training (i.e., success rate = 0 for all episodes), leading to training curves that overlap with the x axis.

Table 3: Parameters of the dynamics model training for local dependency detection experiments. Parameters shared if not specified.

| | Name | Thawing | Tasks CarWash | Kitchen |
|---|---|---|---|---|
| environment | episode length | 20 | 100 | 100 |
| | grid size | 10 | 10 | N/A |
| training | optimizer | | Adam | |
| | learning rate | | $3 \times 10^{-4}$ | |
| | batch size | | 32 | |
| | # of training batches | | 500k | |
| | # of random seeds | | 3 | |
| | mixup Beta parameter | 1 | 1 | N/A |
| ELDEN | activation functions | | ReLU | |
| | $\{\text{MLP}\}_{i=1}^{N}$ | [64, 64] | [64, 64] | [128, 128] |
| | $\lambda$ annealing starts | 50k | 50k | 100k |
| | $\lambda$ annealing ends | 100k | 100k | 200k |
| | # of heads | | 4 | |
| | use bias | | False | |
| | attention   key, query, value size | 16 | 16 | 32 |
| | output size | 64 | 64 | 128 |
| | post attn MLP | [64, 64] | [64, 64] | [128, 128] |
| Input Mask | attention parameters | | same as ELDEN | |
| | $M$ regularization coefficient | | $1 \times 10^{-2}$ | |
| | $M$ regularization annealing starts | 50k | 50k | 100k |
| | $M$ regularization annealing ends | 100k | 100k | 200k |
| Attn Mask | attention parameters | | same as ELDEN | |
| | signature size | | 64 | |
| | SKPMD   learnable bandwidth | | True | |
| | bandwidth initialization | | 1 | |

Table 4: Parameters of the Policy Learning. Parameters shared if not specified.

| | Name | Thawing | Tasks CarWash | Kitchen |
|---|---|---|---|---|
| PPO | optimizer | | Adam | |
| | activation functions | | Tanh | |
| | learning rate | | $1 \times 10^{-4}$ | |
| | batch size | | 32 | |
| | clip ratio | | 0.1 | |
| | MLP size | | [128, 128] | |
| | GAE $\lambda$ | | 0.98 | |
| | target steps | | 250 | |
| | n steps | 60 | 600 | 100 |
| | # of environments | 20 | 20 | 80 |
| training | # of random seeds | | 3 | |
| | intrinsic reward coefficient $\beta$ | | 1 | |
| | # of dynamics update per policy step | | 1 | |
| | dynamics learning rate | | $1 \times 10^{-5}$ | |
| | ensemble size | | 5 | |
| | level of sample prioritization | N/A | N/A | 0.5 |
| | mixup Beta parameter | 0.1 | 0.1 | N/A |
| | partial derivative threshold $\epsilon$ | | $3 \times 10^{-4}$ | |

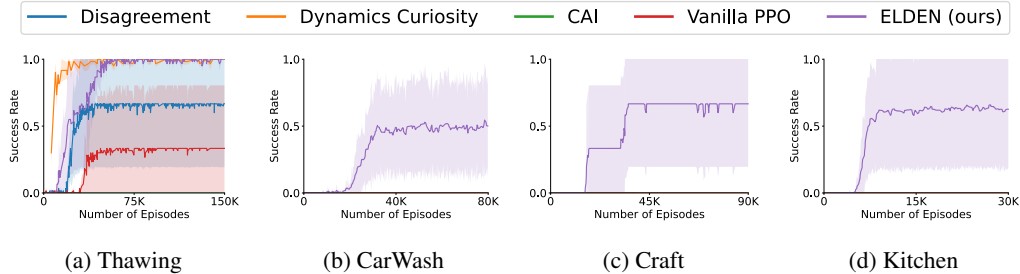

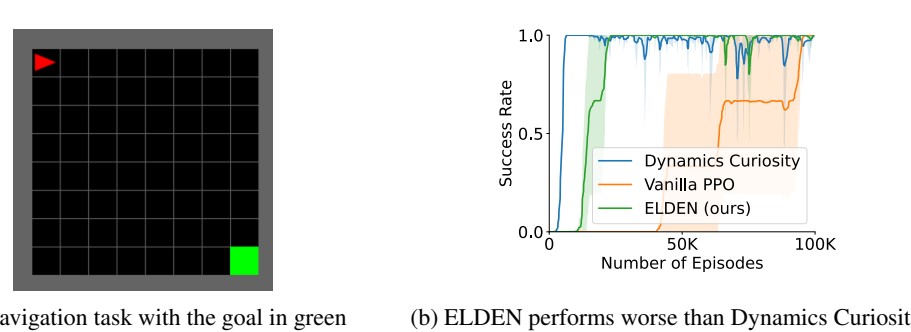

Figure 7: Learning curve of ELDEN (ours) compared to baseline approaches (mean $\pm$ std dev of success rate across three random seeds). For CarWash, Craft, and Kitchen, the success rates for all baselines are zero throughout the training, overlapping with the x axis.

(a) a navigation task with the goal in green      (b) ELDEN performs worse than Dynamics Curiosity

Figure 8: We demonstrate a failure mode of our method on a navigation task.

## E  Failure Modes of ELDEN

As mentioned in the main paper, ELDEN may have limited advantages for tasks that require precise control of a specific environment entity. One such example is navigation, where the agent needs to reach a very specific point in space that has no particular semantic meaning. We empirically examine this statement in the Minigrid environment [7], where the agent needs to navigate to the green goal point in an empty room through primitive actions (turn left, turn right, and move forward), as shown in Fig. 8(a). We compare ELDEN against Dynamics Curiosity and Vanilla PPO, and present the result in Fig. 8(b). Since this environment is relatively simple, all three methods are eventually able to solve the task. However, the Dynamics Curiosity converges faster than ELDEN, showing that ELDEN is indeed not as capable as curiosity-driven explorations in tasks that focus on precise control rather than exploring dependencies between environment entities. The Vanilla PPO converges slowest, indicating that even in the Empty environment, ELDEN still has advantages over purely random exploration.

## F  Compute Architecture

The experiments were conducted on machines of the following configurations:

- Nvidia 2080 Ti GPU; AMD Ryzen Threadripper 3970X 32-Core Processor
- Nvidia A40 GPU; Intel(R) Xeon(R) Gold 6342 CPU @2.80GHz
- Nvidia A100 GPU; Intel(R) Xeon(R) Gold 6342 CPU @2.80GHz

