## A   Neurips Checklist Answer Clarification

During submission, we were not aware of the guidance for filling the checklist and thus misunderstood some questions in it. In this section, we want to correct our answers to some of the checklist questions, if allowed:

- Broader Impacts: n/a

- Experiments: yes, and the code can be found at: `https://github.com/elden-neurips2023/ELDEN`

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

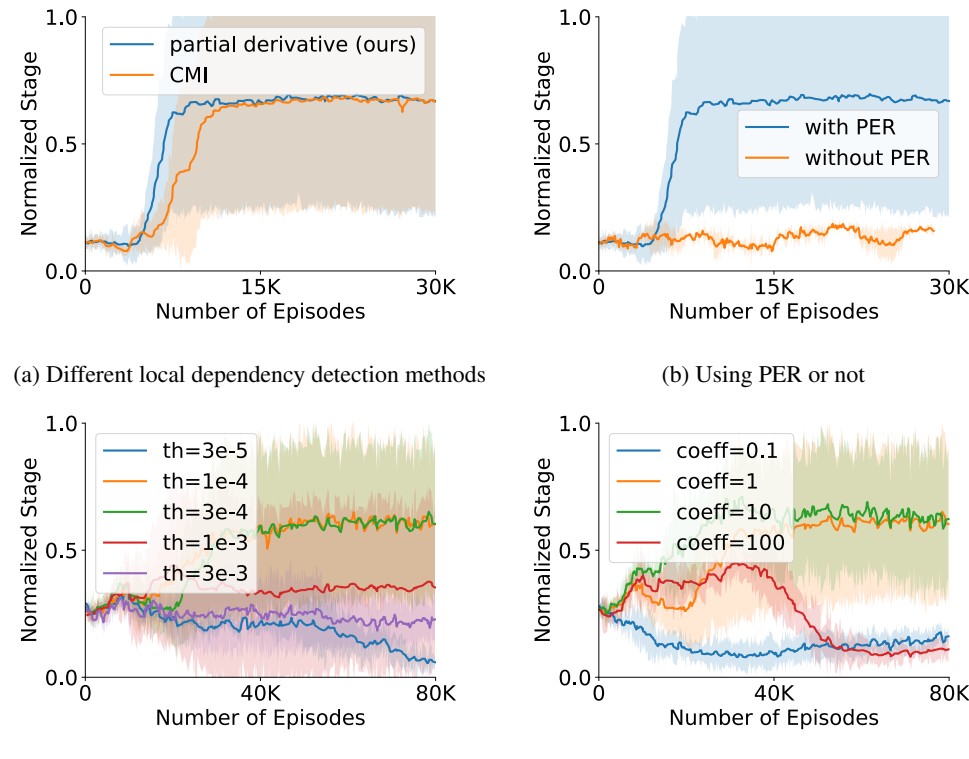

(a) Different local dependency detection methods

(b) Using PER or not

(c) Different thresholds (th) for local dependencies

(d) Different intrinsic reward coefficients

Figure 6: Ablation of ELDEN on task learning.

notice that the computation cost of pCMI is $N$ times more than ELDEN, and thus may not scale to environments with a large number of entities.

### E.3 Ablation of Prioritized Experience Replay

We study the effectiveness of Prioritized Experience Replay (PER) on task learning. Specifically, we test ELDEN with and without PER in the Kitchen environment, and show the result in Fig. 6(b). We can see that ELDEN without PER fails to learn useful policy. The reason is some key entity interactions occur rather rarely before the agent masters them, e.g., there is a small chance for the agent to cook the meatball with random actions. Hence, PER helps the dynamics model learn such infrequent dependencies quickly, enabling it to bias the exploration toward reproducing such dependencies.

### E.4 Ablation of Partial Derivative Threshold

The partial derivative threshold $\epsilon$ determines the dependencies predictions. A threshold that is too large / too small will make all dependency predictions negative / positive respectively, leading to deteriorated performance. In this section, we examine whether our method is sensitive to the choice of threshold in the CarWash environment, where the results are presented in Fig. 6(c). We observe that our method is relatively sensitive to the choice of threshold, and an inappropriate threshold could cause catastrophic failure. A potential next step for ELDEN is to automatically determine the partial derivative threshold.

### E.5 Ablation of Intrinsic Reward Coefficient

The intrinsic reward coefficient controls the scale of the intrinsic reward relative to the task reward. We examine the effect of this coefficient by experimenting with different values in the CarWash environment, where the results are presented in Fig. 6(d). We find that our methods work well in a

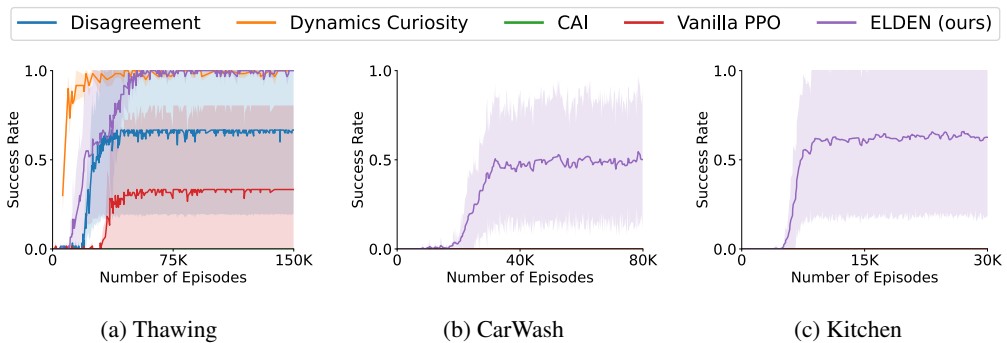

| (a) Thawing | (b) CarWash | (c) Kitchen |

Figure 7: Learning curve of ELDEN (ours) compared to baseline approaches (mean ± std dev of success rate across three random seeds). For both CarWash and Kitchen, the success rates for all baselines are zero throughout the training, overlapping with the x axis.

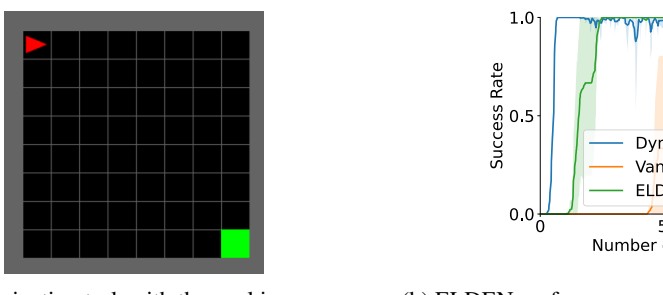

(a) a navigation task with the goal in green     (b) ELDEN performs worse than Dynamics Curiosity

Figure 8: We demonstrate a failure mode of our method on a navigation task.

large range of the intrinsic reward coefficients (1 - 10), since the task only gives sparse rewards and the intrinsic rewards are the only learning signal most of the time. The only exceptions are (1) when the intrinsic reward coefficient is too large (e.g., 100), the intrinsic reward significantly surpasses the task reward, and (2) when the coefficient is too small (e.g., 0.1), the episode intrinsic reward will also be too small (e.g., 0.03) for PPO to learn any useful policy.

### E.6 Success Rate Plots

As a supplementary to the normalized stage metric used in the main paper, we provide the success rate as an additional metric. The success rate learning curves of all methods in the three environments are shown in in Fig. 7. Again, ELDEN outperforms and performs comparably with all baselines. Notice that, in the CarWash and Kitchen environments, all baselines never succeed throughout the training (i.e., success rate = 0 for all episodes), leading to training curves that overlap with the x axis.

## F    Failure Modes of ELDEN

As mentioned in the main paper, ELDEN may have limited advantages for tasks that require precise control of a specific environment entity. One such example is navigation, where the agent needs to reach a very specific point in space that has no particular semantic meaning. We empirically examine this statement in the Minigrid environment [7], where the agent needs to navigate to the green goal point in an empty room through primitive actions (turn left, turn right, and move forward), as shown in Fig. 8(a). We compare ELDEN against Dynamics Curiosity and Vanilla PPO, and present the result in Fig. 8(b). Since this environment is relatively simple, all three methods are eventually able to solve the task. However, the Dynamics Curiosity converges faster than ELDEN, showing that ELDEN is indeed not as capable as curiosity-driven explorations in tasks that focus on precise control rather than exploring dependencies between environment entities. The Vanilla PPO converges slowest, indicating that even in the Empty environment, ELDEN still has advantages over purely random exploration.

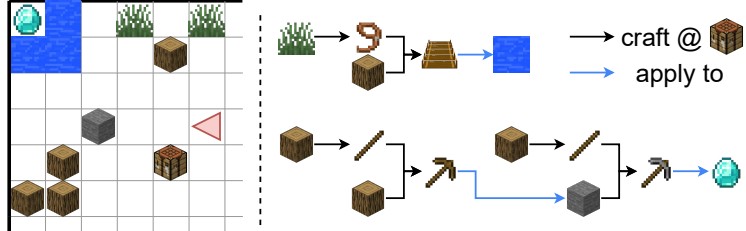

Figure 9: **(Left)** A visualization of the 2-d Minecraft environment. **(Right)** To get the gem, the agent needs to (top) craft a bridge to get across the rive, and (bottom) craft a stone pickaxe that is required to collect the gem.

## G  Minecraft 2D

We also evaluate all exploration methods in a discrete 2-d Minecraft environment that exhibits more objects and more complex entity dependencies than Thawing and CarWash. The environment is modified from the one used by Andreas et al. [1]. Due to limited space in the paper, we chose to show the results in the appendix.

### G.1  Environment Details

As shown in Fig. 9, the environment has complex chained dependencies — to get the gem, the agent needs to

1.  get across the river to reach the gem by

    (a)  collecting a unit of grass and crafting a rope,

    (b)  collecting a unit of wood and crafting a bridge with the rope,

    (c)  building the bridge on top of the river;

2.  collect the gem by

    (a)  collecting a unit of wood to craft a wood stick

    (b)  collecting another unit of wood and combining it with the stick to craft a wood pickaxe that is required for collecting the stone,

    (c)  collecting a unit of wood and a unit of stone to craft a stick and then a stone pickaxe that is required for collecting the gem,

    (d)  collecting the gem with the stone pickaxe.

Notice that all crafting must be conducted at the crafting table. The discrete state space consists of (i) the agent's

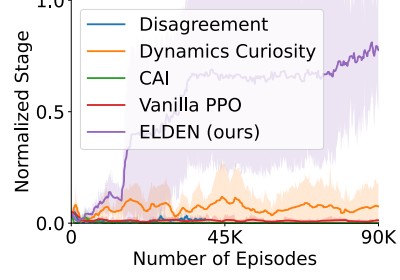

Figure 10: Learning curve of ELDEN (ours) compared to baseline approaches (mean $\pm$ std dev of the number of stages completed across three random seeds) in the 2-d Minecraft environment.

position and direction, (ii) an inventory tracking the number of materials and tools that the agent has, and (iii) the positions of all environment entities. The discrete action consists of (i) picking up / applying tools to (only effective when the agent faces an environment entity and has the necessary tools to interact with it), (ii) crafting a specified tool (only effective when the agent has enough materials and faces the crafting table), and (iii) moving to a specified environment entity.

### G.2  Evaluating Exploration in 2-d Minecraft with Sparse Rewards

As shown in Fig. 10, even though the task requires that the agent follows the complex craft procedure with complex chained dependencies, ELDEN still learns to finish the task. In contrast, other exploration method fails to finish the task, only ending up with crafting one or two useful tools. This result again demonstrates ELDEN's advantage in exploring complex interactions between a large number of environment entities.

## H Compute Architecture

The experiments were conducted on machines of the following configurations:

- Nvidia 2080 Ti GPU; AMD Ryzen Threadripper 3970X 32-Core Processor
- Nvidia A40 GPU; Intel(R) Xeon(R) Gold 6342 CPU @2.80GHz
- Nvidia A100 GPU; Intel(R) Xeon(R) Gold 6342 CPU @2.80GHz