# OpenReview forum: "ELDEN: Exploration via Local Dependencies"
_NeurIPS.cc/2023/Conference — NeurIPS 2023 poster_

### Official Review · Reviewer_xY6d · 2023-06-27

**Soundness:** 3 good
**Presentation:** 2 fair
**Contribution:** 2 fair
**Rating:** 5
**Confidence:** 4

**Summary:**

This work proposes `ELDEN, Exploration via Local DepENdencies`, a framework with an intrinsic reward that encourages the discovery of new interactions between entities such as the agent or objects that have some influence on each other. The method uses partial derivative of the learned dynamics to model the local dependencies between entities. The uncertainty of the predicted dependencies is used as an intrinsic reward. What distinguishes this from other related work, is that the focus for defining what makes a state interesting. Traditional approaches may focus on the specifics of how entities in an environment interact. In contrast, here interesting states are defined
based on whether the entities *can* interact, without focusing on the specifics of how the interaction occurs. Specifically, the algorithm is biased towards exploring states where the relationships between entities are not well understood.

To implement this, the authors train an ensemble of dynamics models. Within each model in the ensemble, local dependencies between objects are modeled using partial derivatives of state predictions with respect to the current state and action. Partial derivatives can capture how changes in state or action affect changes in predictions, which can be thought of as capturing local interactions. The uncertainty in local dependencies is then quantified as the variance across all the models in the ensemble where a high variance suggests high uncertainty.

For learning the object interactions themselves, they propose using a `Causal Graphical Model` to represent the transition at a given time step 't'. CGM consists of three parts: `1. Nodes`: These represent the state at time step 't' (S_t), the action at time step 't' (A_t), and the state at the next time step 't+1' (S_t+1). `2. Directed Graph G`: This graph illustrates the global dependencies between the nodes.
`3. Conditional Distribution 'p'`: For each state variable at the next time step (S_n_t+1),  there is a conditional distribution 'p' which represents the probability of reaching that state given the current state and action.

The assumption is that the transition probability `P(s_t+1 | st, at)` can be factorized into a product of conditional probabilities for each state variable at the next time step. Here, `Pa(v)` refers to the parent nodes of a node `v` in the causal graph `G`. This means that the probability of transitioning to a new state depends on the current state and action, and can be represented as a product of several smaller probabilities based on the causal relationships.

Instead of using these global graph `G` which includes all possible dependencies, the authors use a local causal graph `G_t`.
This Local Causal Graph Model is aimed at more accurately and efficiently representing the real-world scenarios by focusing on relevant
dependencies at a given time rather than considering all possible interactions. This can potentially lead to more efficient learning and better generalization in reinforcement learning environments.

ELDEN's main goal is to identify the dependencies that are locally active among the entities in the environment.
In other words, for a specific state 'st' and action 'at',
ELDEN tries to determine which entities in the environment have an active causal relationship with each other.

This involves constructing a graph where the nodes represent entities, and the edges represent active causal relationships between them.
This graph is specific to the given state and action (st, at), and is called the local causal graph 'Gt'.

 **Experiments**

Experiments are conducted with the following baselines: `pCMI (point-wise conditional mutual information)`,  `Attn (attention)`: Use the score between each entity pair computed by the attention modules inside the dynamics model to quantify local dependencies. `Vanilla PPO`.

Presented tasks : `Kitchen Task` has a very high variance! ` Thawing Dynamics` curiosoty outperorms ELDEN. `CarWash` results also has high variance.



**Strengths:**

- The ideas presented here are very interesting and novel. Creating an intrinsic reward based on local dependency, which aids exploration in RL, especially in environments with sparse rewards is novel.

**Weaknesses:**

- Weak Experimental results. The results as they stand are not very convincing.
That being said, these are hard problems so I understand the challenges in making these environments to work.

**Questions:**

- For the PPO baseline this serves as a baseline with sparse reward? So it is not surprising that this baseline doesnt really work ?

**Limitations:**

- Underwhelming experiments.

---

> ### Author Rebuttal · Authors · 2023-08-10
>
> We thank the reviewer for the detailed reading of our paper and for the very constructive suggestions! We hope our responses adequately address the following concerns regarding the evaluation of our work.
>
> >The results as they stand are not very convincing. That being said, these are hard problems so I understand the challenges in making these environments work.
>
> While we appreciate where the reviewer is coming from, we submit that the experimental results are stronger than they may appear to be on the surface. Specifically,
>
> - High variance is normal: Notice that under our problem setup, the agent gets a reward only if it successfully completes the task. For such tasks, it is very common for successful exploration methods to have high variance. (for example, in Fig 5 (c) of Pathak et al [1]). Intuitively, this is because the intrinsic reward can at its best provide some sort of state-coverage guidance, and the agent often has to rely on luck to obtain its first successful run, which may never happen for some random seeds.
> - Extra experiment domains: Also notice that we provide extra experiments on the crafter (Minecraft 2D) domain in appendix Sec G. In this challenging environment, the agent has > 20 objects to interact with, 5 tools that it can craft, and a complex technology tree to follow, where the agent has to efficiently explore through these interactions.
> - Additional baseline: Additionally, as part of the rebuttal, we added new experiments that compare against a new baseline, RND, on the crafter domain. Results can be seen in the global response in Fig 1.
>
> [1] Deepak Pathak, Pulkit Agrawal, Alexei A. Efros and Trevor Darrell. Curiosity-driven Exploration by Self-supervised Prediction. In ICML 2017.
>
> > For the PPO baseline this serves as a baseline with sparse reward? So it is not surprising that this baseline doesnt really work?
>
> Yes, as described in Sec 4.2 Vanilla PPO is a baseline that learns from sparse reward only.  So as the reviewer points out, it is not expected to work in the challenging settings that we consider.

---

> > ### Comment · Reviewer_xY6d · 2023-08-13
> > **Response to authors**
> >
> > I acknowledge that I have read the reviews and I have no further questions.

---

### Official Review · Reviewer_G2vP · 2023-07-03

**Soundness:** 2 fair
**Presentation:** 3 good
**Contribution:** 2 fair
**Rating:** 5
**Confidence:** 4

**Summary:**

The paper proposes an intrinsic exploration reward that explicitly takes dependencies between different entities in the environment into account. Such an intrinsic reward helps sparse-reward RL in a few skill-based domains.

**Strengths:**

- I find the idea of modeling dependency graphs between entities interesting and worth exploring.
- The related work section is written especially well, making it very easy to understand how the proposed work is contextualized.
- Explanation of the method is clearly written and easy to understand.
- Results on given domains are promising.

**Weaknesses:**

- The major weakness of the paper is in the evaluation. In particular, the domains that are used for evaluation all use high-level primitives (like goto object and pickup object) which results in relatively few steps in order to accomplish a goal. It's hard to argue that this setting is comparable to the difficulty of something like Franka Kitchen (low-level control in a kitchen environment with sparse reward). And especially given many moving parts are required to make this work, it would limit the applicability in my reading. It's even more unfortunate that this is not justified and only discussed in detail in Appendix C, given that it transforms a long-horizon sparse-reward problem (the typical understanding) into a relatively short-horizon problem.
- With regards to the above, ELDEN is proposed as a standalone method, but in the most complex domain tested (Kitchen), an additional exploration method, PER, is required in order to make it work.
- Too much is pushed to the appendix, for instance architectural details that are important to understand for evaluation (Attn Mask in section 4.1) are only discussed in Figure 4. Ablations for a method with many moving parts only show up in Appendix E.
- All domains are discrete action, and the most convincing experiments (no PER required) are discrete state. The use of partial derivatives through Mixup is a little bit strange and I don't understand where the claim "dynamics models trained on such data... reflect local dependencies" is justified.

**Questions:**

- How is the variance of the dependency graph computed (Algorithm 1)?
- Is there a domain with low-level control in which ELDEN can be benchmarked?

**Limitations:**

Some limitations are addressed in the Conclusion, and more interesting ones are also discussed in Appendix F. In particular Appendix F makes the point that ELDEN is suboptimal in navigation environments, and thus that it is particularly scoped for settings with a big dependency graph.

---

> ### Author Rebuttal · Authors · 2023-08-10
>
> We thank the reviewer for the detailed reading of our paper and constructive suggestions! We hope our responses adequately address the following concerns regarding the evaluation of our work.
>
> > The experiment domains use high-level primitives which results in relatively few steps in order to accomplish a goal. It's hard to argue that this setting is comparable to the difficulty of something like Franka Kitchen (low-level control in a kitchen environment with sparse reward).
> (from the questions section: Is there a domain with low-level control in which ELDEN can be benchmarked?)
>
> We agree with the reviewer that the Franka Kitchen environment is interesting and challenging. However, we believe environments with high-level primitives that focus on a different level of exploration challenges are not inherently less interesting or easier to solve. Specifically,
>  - As mentioned in the introduction, our method aims to solve tasks with many interaction modes, where the challenge is to **find the correct sequence of interactions among many options** that leads to task success. For example, in our crafter (Minecraft 2D) environment described in appendix G, the agent has > 20 objects to interact with and 5 tools that can craft, and the agent has to efficiently explore through these interactions. As demonstrated by our experiments, even with high-level primitives, all of the baseline methods fail to solve the task (except for thawing), establishing that dealing with many interaction modes is indeed challenging.
>  - By comparison, the original Franka Kitchen environment [1] has almost no object dependencies, where each object can be toggled independently and the task objective is just to achieve multiple independent goals, one for each object. As such, low-level control in the Franka Kitchen environment is challenging in the aspect of **goal-reaching** (moving the end-effector to the object, similar to navigation) and **precise sensorimotor control** (grasping the object), but not in the aspect of selecting the correct interaction among many options, making it orthogonal to the focus of our method.
>
> That being said, we provide new experimental results examining the performance of ELDEN on low-level, continuous domains in the global response. Specifically, we test replacing the action primitives in our kitchen environment with low-level position control of the gripper.
> As shown in Fig 3 in the global response, the problem becomes so hard that none of the methods can make meaningful progress. To the best of our knowledge, most methods that can learn with low-level actions and sparse rewards require some form of human priors (like offline data) [2].
>
> We agree with the reviewer that an exciting future direction of this work would be to combine ELDEN with more fine-grained exploration methods to solve domains that entail both rich interaction and low-level controls.
>
> [1] Gupta, Abhishek, et al. Relay policy learning: Solving long-horizon tasks via imitation and reinforcement learning. CoRL, 2019.
>
> [2] Alakuijala, Minttu, et al. Learning reward functions for robotic manipulation by observing humans. ICRA 2023
>
> >And especially given many moving parts are required to make this work, it would limit the applicability in my reading. With regards to the above, ELDEN is proposed as a standalone method, but in Kitchen domain, an additional exploration method, PER, is required in order to make it work.
>
> We want to clarify, to generate the intrinsic reward, our method only needs to learn an ensemble of dynamics models.
>
> Please note that prioritized experience replay (PER), or sample prioritization, is applied to **dynamics training** only, rather than an exploration method for policy learning. As the data buffer grows with collected transitions, PER is a common technique to let the dynamics model learn from new data efficiently, and it is widely used in other intrinsic reward methods, like Lobel et al [3].
>
> For all methods, we use PPO to learn policy. PPO is an on-policy algorithm that does not learn from the replay buffer and **PER is not applicable**.
>
> [3] Lobel, Sam, Akhil Bagaria, and George Konidaris. Flipping Coins to Estimate Pseudocounts for Exploration in Reinforcement Learning. ICML 2023
>
> >Too much is pushed to the appendix, for instance architectural details that are important to understand for evaluation (Attn Mask in section 4.1) are only discussed in Figure 4. Ablations for a method with many moving parts only show up in Appendix E.
>
> We agree with the reviewer that details and justifications of the environments, as well as some ablation experiments should be moved to the main text. We will prioritize making space for that in the next version of our paper.
> However, with regards to Attn Mask, it is a baseline that we describe clearly in the text of Section 4.1. The architecture is not our own contribution, so we consider it appropriate to appear only in the appendix.
>
> >The use of partial derivatives through Mixup is a little bit strange and I don't understand where the claim "dynamics models trained on such data... reflect local dependencies" is justified.
>
> Mixup has been shown to smooth the partial derivatives of the model output w.r.t. inputs (Fig 1 (b) and Fig 2 (b) in [4]), and thus is suitable to improve partial derivative estimation in our method.
> For the justification that mixup improves local dependency identification, the results are shown in Table 3 (the “no Mixup” row vs others).
>
> [4] Zhang, Hongyi, et al. "mixup: Beyond empirical risk minimization." arXiv preprint arXiv:1710.09412 (2017).
>
> >How is the variance of the dependency graph computed?
>
> Thanks for pointing this out. Each dependency graph is represented as a binary adjacency matrix. Given a set of graphs, we calculated their per-entry variances and used the mean acorss all entries in the graph as the eventual variance. We will include this specification in the next version of the paper.

---

> > ### Comment · Reviewer_G2vP · 2023-08-14
> >
> > Thanks to the authors for their many clarifications. I have some specific responses below.
> >
> > - **Low-level control + ELDEN**: I provided Franka Kitchen as an example given that it's close to the authors tested tasks as there is a dependence to the objects in that reward is only obtained when the objects are interacted with in the correct sequence, not independently. Still, I appreciate the additional experiment and clarification. I do think that the current tasks are a bit toy given that they rely on such a structured action space, but I appreciate the paper's perspective and the failure of other exploration methods a bit more.
> > - **Moving parts**: I had a misunderstanding in my initial read as to how PER is integrated and I thank the authors for clearing this up.
> > - **Too much in appendix**: Thanks for the clarification on Attn Mask, I agree that it probably makes sense to leave in the appendix.
> > - **Partial derivatives**: Thanks for the clarification, without familiarity with these precise plots it is difficult to find which prior results the paper refers to, and it'd be nice to have a reference to Table 3 in the appendix here.
> > - **Variance of graph**: Thanks for this clarification
> >
> > I think the paper is stronger than I had initially judged, though I still believe that the tasks tested are somewhat toy. In particular, for the simulated Kitchen, which is presented as the most "real" of the tasks, due to its reliance on exact positions of objects and grasping primitives. I do think that I would be happy to update my score to a 5 though.

---

> > > ### Author Response · Authors · 2023-08-15
> > >
> > > We thank the reviewer for the thoughtful feedback. We are glad to see that our responses addressed the concerns raised by the reviewer and will make sure to include all suggestions from the reviewer and new analyses in the paper.
> > >
> > > We thank the reviewer for agreeing to increase their score to 5. We noticed that at the moment the rating for our paper is still 4, and would like to gently remind the reviewer about this potential oversight.

---

> > > > ### Comment · Reviewer_G2vP · 2023-08-15
> > > >
> > > > Thanks for the reminder on score-updating, I wanted to see if there were any additional thoughts on the authors' side first.

---

### Official Review · Reviewer_FZmh · 2023-07-05

**Soundness:** 3 good
**Presentation:** 4 excellent
**Contribution:** 3 good
**Rating:** 5
**Confidence:** 3

**Summary:**

This paper proposes a novel intrinsic reward called ELDEN for reinforcement learning. ELDEN encourages the discovery of new local dependencies between entities. Experiments on some robotic tasks are carried out to valid the idea.

**Strengths:**

1. The idea is novel and interesting.

2. The usage of dynamics models to approximate causal dependency is interesting.

3. The method does not make any assumption of knowing groundtruth dependency.

4. The paper is well-written. The authors discussed their method are different from existing works.

**Weaknesses:**

1. My concern is about the significance of proposed method. This work focuses on solving semantic-level tasks and assumes knowing underlying objects. Then, why should we use RL in this case? A better option is probably using large language models (LLM) to do the semantic reasoning directly. Another option is to use LLM to design exploration reward like [1]. I would like to see authors opnion on this.

2. In the experiments, two of the considered tasks are not solved (reach normalize score 1.0). The performance is not strong enough.

3. The authors can compare to more baselines. One important baseline can be RND.

[1] Yuqing Du, Olivia Watkins, Zihan Wang, Cedric Colas, Trevor Darrell, Pieter Abbeel, Abhishek Gupta, Jacob Andreas. Guiding Pretraining in Reinforcement Learning with Large Language Models. In ICML 2023.

**Questions:**

1. What are the state-space and action-space of the used tasks (including the meanings and number of dimensions)? How long is the horizon for each task?

2. How does proposed method perform in image-based tasks?


**Limitations:**

The authors discussed their limitations and potential negative societal impact properly.

---

> ### Author Rebuttal · Authors · 2023-08-10
>
> We thank the reviewer for the detailed reading of our paper and constructive suggestions! We hope our responses adequately address the following concerns regarding the significance and evaluation of our work.
>
> > Weakness 1: compare ELDEN (and RL in general) with LLMs
>
> Thank you for making this point. While we are as excited as everyone else about the possibilities of leveraging LLMs in robotics research, we also think it’s important that the community does not lose interest in research outside the scope of LLMs.  While it may indeed be possible to study a similar problem through the lens of LLMs, please note that
> **ELDEN does not rely at all on the semantic meaning of state factors and actions**. It only requires that the state space is factored. As pointed out by R1, this is itself an important assumption to examine.  But it is a completely separate assumption than those made by LLM-based research.Specifically, ELDEN differs from research using LLMs (including Du et.al 2023 [1]) in at least the following ways:
> - LLMs require the semantic meaning of each state factor and action for effective reasoning.
> - Using LLMs for reasoning assumes that LLMs know how the environment transitions, but many environments are naturally hard to describe in language and are thus not well learned by LLMs. In contrast, our method (and RL in general) does not assume pre-existing knowledge of the transition probabilities and can learn from the environment interactions.
> - Using LLMs constrains the performance to be at the human level, while our method (and RL in general) has the potential to surpass humans and find novel strategies for solving the task, like how AlphaGo finds new Go strategies.
>
> >Weakness 2: two tasks are not solved.
>
> Please note that in those two tasks, our method was able to solve them for 2 out of 3 random seeds. The other seed only achieves some intermediate stages, rendering the average score smaller than 1.0.
> For the unsuccessful seed, notice that under our problem setup, the agent gets a reward only if it successfully completes the task. **For such tasks, it is very common for successful exploration methods to have high variance.** (for example, in Fig 5 (c) of Pathak et al [2]). Intuitively, this is because the intrinsic reward can at its best provide some sort of state-coverage guidance, and the agent often has to rely on luck to obtain its first successful run, which naturally introduces lots of stochasticity.
>
> [2] Deepak Pathak, et al. Curiosity-driven Exploration by Self-supervised Prediction. In ICML 2017.
>
> >comparison with RND.
>
> In the global response, we present new experiments comparing against RND on the challenging Minecraft 2D domain, where our method significantly outperforms all baselines including RND.
>
> >Environment specifications (state and action space, their meanings, and horizon)
>
> We list the environment specifications below and will include the extra specifications in the next version of the paper (note that the **MEANINGS** of state variables and actions are not given to our method). Given the space limits, we put the specification of Thawing and CarWash environments in the second part of the global response:
> - Kitchen:
>   - state space (43d), consisting of robot end-effector position (3d) and velocity (3d), robot gripper angles (2d) and angular velocity (2d), butter position (3d) and quaternion (4d), butter melting state (1d), meatball position (3d) and quaternion (4d), meatball cook state (1d), pot position (3d) and quaternion (4d), stove position (3d), target position (3d),  stove switch position (3d) and state (1d),
>   - action space (12d), consisting of
>     - move the hand to 1) the butter, 2) the meatball, 3) above the pot, 4) the pot handle, 5) the stove, 6) the stove switch.
>     - grasp 7) the butter, 8) the meatball, 9) the pot handle, if the hand is close to it
>     - 10) turn on/off the stove switch, if the hand is close to it
>     - 11) open the gripper
>     - 12) no action, which is useful for waiting for the meatball to be cooked
>   - given horizon: 200
>   - minimal # of time steps required to finish the task: 25
> - Minecraft 2D:
>   - state space (40d), consisting of
>     - agent position (2d) and direction (1d)
>     - positions (2d) of the 1 gem block, 5 water blocks, 2 grass blocks, 5 wood blocks, and 1 stone block (28d in total).
>     - the number of materials (grass, wood, stone, block) and tools (axe stick, wood axe, stone axe, rope, bridge) in the inventory (9d).
>   - action space (21d):
>     - move next to each of the environment objects (gem, water, grass, wood, stone blocks, and the workbench), if there is a path (15d in total)
>     - craft a tool among the choice of (axe stick, wood axe, stone axe, rope, bridge) if the agent has enough materials and is next to the workbench (5d)
> collect the material that faced by the agent, applicable only if the agent has the tool necessary for collection (1d)
>   - given horizon: 200
>   - minimal # of time steps required to finish the task: 23
>
> In these environments, the exploration challenges are caused by the large action space and the chained dependencies — the agent needs to efficiently explore throughout many interaction modes and find the correct order of interactions that leads to task success.
> For all environments, the actions only have effects when their preconditions are met, i.e., in the kitchen environment, grasping the butter will execute only if the robot hand is empty and close to the butter.
>
> >How does the proposed method perform in image-based tasks?
>
> Solving image-based tasks is beyond the scope of this paper. However, a simple potential way to extend our method to image-based tasks is to extract disentangled features from an image (e.g. Schölkopf, Bernhard, et al. [1]) and use the extracted features as the state space.  We will add this as an interesting direction for future work.
>
> [1] Schölkopf, Bernhard, et al. "Toward causal representation learning." Proceedings of the IEEE 109.5 (2021): 612-634.

---

> > ### Comment · Reviewer_FZmh · 2023-08-10
> > **Thank you for the response.**
> >
> > I would like to thank authors for the detailed response. For weakness 1, the authors point out that ``many environments are naturally hard to describe in language and are thus not well learned by LLMs''. I agree with this, but my concern is that if ELDEN is able to solve this kind of problem. For example, it is hard for LLM to solve fine-grained, low-level robotic locomotion or manipulation problems. However, ELDEN could not solve this problem effectively either (e.g. the updated cheetah results).
> >
> > In conclusion, I would like to know if there is any specific task that ELDEN can solve but LLM and other generic exploration methods fail to solve.

---

> > > ### Author Response · Authors · 2023-08-11
> > >
> > > Thank you for your quick response!
> > >
> > > Firstly, we want to point out that we do compare against other generic exploration methods in Sec 4 and Appendix G of our paper, where ELDEN consistently outperforms them across all of our evaluation domains. These are clearly tasks that “ELDEN can solve but other generic exploration methods fail to solve”.
> > >
> > > With respect to “tasks that ELDEN can solve and LLMs fail to solve”: We want to emphasize that using LLM for decision-making/exploration has a lot more limitations than just the inability to handle low-level controls, including but not limited to that **LLMs require the semantic meaning of actions and state variables, and environment-specific prompt engineering** [1].
> > > - With regard to actions: For example, if the high-level skills in our kitchen environment are learned through data-driven methods such as unsupervised skill discovery methods [2][3], it will be hard to describe such behavior in language, causing LLM-based methods to fail.
> > > - With regard to state variables: For example, Minecraft keeps updating its contents, such as new tools and enemies, and LLMs do not know how those tools can be used nor how new enemies behave.
> > >
> > > In both cases, ELDEN can be directly applied to solve the problem, as in our experiments, since it does not require knowing the semantic meaning of these states and actions.
> > >
> > > We share your enthusiasm for the possibilities that LLMs represent for the future of machine learning and robotics. However, from our perspective, research on using LLMs to aid exploration is still in its infancy and there are no established benchmarks that we can easily compare against.
> > >
> > > We are concerned that exclusively requiring papers to directly compare with LLMs could potentially have a negative impact on the field, especially on tasks where LLMs' effectiveness has not been demonstrated without additional assumptions (e.g., semantic meanings, environment-specific prompt engineering).
> > >
> > > If the reviewer has a concrete suggestion for an available system that can accomplish the tasks demonstrated in this paper, we would be delighted to try it out and compare it with our method.
> > >
> > > [1] Yuqing Du, Olivia Watkins, Zihan Wang, Cedric Colas, Trevor Darrell, Pieter Abbeel, Abhishek Gupta, Jacob Andreas. Guiding Pretraining in Reinforcement Learning with Large Language Models. In ICML 2023. \
> > > [2] Eysenbach, Benjamin, et al. "Diversity is all you need: Learning skills without a reward function." arXiv preprint arXiv:1802.06070 (2018). \
> > > [3] Groth, Oliver, et al. "Is curiosity all you need? on the utility of emergent behaviors from curious exploration." arXiv preprint arXiv:2109.08603 (2021).

---

> > > > ### Comment · Reviewer_FZmh · 2023-08-11
> > > > **Thank you for your response**
> > > >
> > > > Thank you again for the response. Although I still remain a bit uncertain and skeptical, I appreciate many clever ideas presented in the paper and authors' engineering effort. Therefore, I decide to raise my score to borderline accept. I encourage the authors to look into ELDEN's practical applications, and study its integration with other modern ML components to build a truly useful agent in the future.

---

### Official Review · Reviewer_kab1 · 2023-07-06

**Soundness:** 4 excellent
**Presentation:** 4 excellent
**Contribution:** 3 good
**Rating:** 7
**Confidence:** 4

**Summary:**

This paper proposes ELDEN, a method for intrinsic reward based on local dynamics dependencies between factored state variables. It learns an ensemble of factored dynamics models, and uses the magnitude of the partial derivatives to detect local dependencies between state variables, and uses the ensemble variance as an exploratory intrinsic reward. This method is tested across a variety of environments such as gridworlds and robotic control tasks, showing great accuracy in capturing local dependencies and exploratory performance.

**Strengths:**

This paper presents a very clear and thorough test of the main ELDEN method. ELDEN itself is straightforward to understand, and section 4.1 does a convincing job of evaluating how well it detects local dependencies compared to reasonable baselines. Section 4.2 also has reasonable baselines, and clearly demonstrates the effectiveness of ELDEN. The ablations on mixup and the regularization are also well done.

Overall, many interesting questions around ELDEN are covered by the paper, and the results are very clear.


**Weaknesses:**

A common weakness, not specific to ELDEN, is of course the assumption of the factored state space. While this can be overcome with object-identifying representations, there is still the question of how effective ELDEN may be when paired with learned factored representations as opposed to ground truth. Hopefully future work can look in this direction.

While it is mentioned that ELDEN is not meant to excel in all domains, an additional experiment in a more common domain such as the sparse reward tasks of DM Control Suite would be enlightening to see. Perhaps ELDEN would also work well, or it may not compared with the baselines; either way it would provide some additional insight into the limitations of ELDEN.

---- After Author Rebuttals ----

After reading other reviews and author rebuttals, I maintain my score. I think this paper has clearly shown when and where ELDEN can excel, and the limitations when ELDEN is not very effective.

**Questions:**

It is mentioned that ELDEN can help with more complex, chained dependencies, however the basic method is still only looking at 1-step local dependencies. Why would one expect ELDEN to do better than other 1-step methods?

**Limitations:**

Are addressed appropriately in the paper.

---

> ### Author Rebuttal · Authors · 2023-08-10
>
> We thank the reviewer for the detailed reading of our paper and constructive suggestions! We hope our responses adequately address the following questions about our work.
>
> > A common weakness, not specific to ELDEN, is of course the assumption of the factored state space. Hopefully future work can look in this direction.
>
> We agree that testing the effectiveness of ELDEN when the factored representations are learned would be interesting next steps.
>
> > While it is mentioned that ELDEN is not meant to excel in all domains, an additional experiment in a more common domain such as the sparse reward tasks of DM Control Suite would be enlightening to see. Perhaps ELDEN would also work well, or it may not compared with the baselines; either way it would provide some additional insight into the limitations of ELDEN.
>
> In the global response, we add the results of the DMC Cheetah domain with sparse rewards in Fig 2. As expected by the reviewer, ELDEN and empowerment method (CAI) performs worse than the curiosity-based method. We analyze the reasons as follows:
>
> - With regard to ELDEN, DMC with sparse rewards is challenging in the aspect of precise low-level sensorimotor control, which is orthogonal to the focus of ELDEN on selecting the correct interaction among many options. Hence, it is not surprising that ELDEN does not help with the exploration in the cheetah domain.
> - With regard to CAI: the cheetah task requires precise control of each joint. In contrast, CAI is motivated to maximize its control over all joints (rotating them between the joint limits), and thus it fails to learn the tasks and harms the exploration.
>
> > It is mentioned that ELDEN can help with more complex, chained dependencies, however, the basic method is still only looking at 1-step local dependencies. Why would one expect ELDEN to do better than other 1-step methods?
>
> Take the Minecraft 2D environment in Appendix G as an example, where it has the chained dependencies of “collect wood” $\rightarrow$ “craft wood axe” $\rightarrow$ “collect stone” $\rightarrow$ “craft stone axe” $\rightarrow$ “collect gem”. As shown in Appendix G, our method solves this problem better compared to baselines (dynamics curiosity, dynamics uncertainty, CAI, and RND) because:
> - Our method prioritizes novel 1-step interactions (local dependencies). In the initial learning stage, compared to baselines that are preoccupied with repetitive movements for seeking novel state values (e.g., new agent positions after movements), our approach prioritizes novel interactions (e.g., collecting wood, grass, or other objects). This proactive exploration strategy increases the agent's chances of successfully collecting wood, which, in turn, **opens up new possibilities for further interactions**, such as crafting the wood axe.
> - Though the reward is based on 1-step local dependencies, as RL optimizes the return, the agent will seek both current and future novel dependencies. For example, as the agent undergoes training, it may become familiar with former interactions in the chain (i.e., up to “collect stone”). However, it remains motivated to tackle the latter, less familiar dependencies that follow the “craft stone axe”. Hence, even if the former dependencies do not give any 1-step reward, the agent will still finish them to continue exploring the latter dependencies and maximize its return.
>
> We will include this discussion in the next version of the paper.

---

> > ### Comment · Reviewer_kab1 · 2023-08-10
> > **Clarifying the 1-step vs chained dependencies**
> >
> > Thank you for addressing my questions!
> >
> > I want to clarify what I meant by asking "Why would one expect ELDEN to do better than other 1-step methods?". This was in part motivated by the following statement about empowerment in Section 2.2: "However, due to the difficulty in measuring the mutual information across a multi-step trajectory, existing empowerment-based methods only measure 1-step empowerment, ...", but also about the repeated emphasis on chained dependencies. The two mechanisms you mentioned in your response are general properties of RL, which would equally apply to other methods such as 1-step empowerment or 1-step curiosity - the RL itself will learn to visit parts of the state space with large aggregate novelty. The fact that "chained dependencies" is mentioned so many times in the paper could give the wrong impression that ELDEN is trying to learn a much more complicated dependency graph, whereas it is actually a 1-step method just like many other methods. So I think it could be useful to just add a bit of clarity that ELDEN is still a 1-step model, but is able to handle chained dependencies because of RL.

---

> > > ### Author Response · Authors · 2023-08-11
> > > **Thanks for your suggestions on improving clarity!**
> > >
> > > Thank you for your quick response! We agree that the part about "chained dependencies" should be clearer, and we will add the suggested clarification in the paper.

---

> > > > ### Comment · Reviewer_kab1 · 2023-08-14
> > > >
> > > > Thank you also for the Cheetah experiment. I think it's good to make it clear when ELDEN excels, and its limitations. Thus I will maintain my score.

---

### Author Rebuttal · Authors · 2023-08-10

We thank all reviewers for the detailed reading of our paper and constructive suggestions! In the global response, we would like to describe the setup of additional experiments and the results can be found in the attached pdf.
- Comparison with RND: Using the challenging crafter (Minecraft 2D) domain described in Appendix G (featuring > 20 objects for the agent to interact with and 5 tools to craft, with a complex technology tree), we compare our method against RND. As shown in Fig 1, our method ELDEN significantly outperforms all baselines, which again demonstrates the strength of ELDEN in solving tasks with many interaction modes and complex preconditions.
- Results on a deepmind control suite (DMC) domain, asked by R1 kab1: In Fig 2, we show the performance in the Cheetah environment with sparse reward. Following the "no free lunch theorem", no intrinsic reward method can perform best in all environments. As discussed in Sec 5 (and above), our method ELDEN aims to solve tasks with many interaction modes, where the challenge is to find the correct interaction leading to the task success. However, DMC with sparse rewards is challenging in the aspect of precise low-level sensorimotor control. Hence, unsurprisingly, ELDEN performs worse than curiosity-based methods.
- Results on using low-level actions in the Kitchen domain, asked by R3 G2vP: In Fig 3, in the kitchen environment, we use low-level actions consisting of end-effector x, y, z movements (in the range of [-5, 5] cm) and gripper control (in the range of [-1, 1], where > 0 for closing the gripper and < 0 for opening), which is one of the low-level action formats provided in the robosuite [1]. Similar to DMC domains, using low-level actions is challenging in the aspect of goal-reaching (moving the end-effector to the object, similar to navigation) and precise sensorimotor control (grasping the object). To the best of our knowledge, in such manipulation domains, most methods that can learn from sparse rewards with low-level actions require some form of human priors (like offline data) [2][3]. As a result, none of the methods make meaningful progress.

With regard to DMC domains and manipulation domains with low-level actions, we agree with the reviewer that an exciting future direction of this work would be to combine ELDEN with more fine-grained exploration methods to solve such domains that entail both rich interaction and low-level controls.

[1] Zhu, Yuke, et al. "robosuite: A modular simulation framework and benchmark for robot learning." arXiv 2020.
[2] Gupta, Abhishek, et al. Relay policy learning: Solving long-horizon tasks via imitation and reinforcement learning. In CoRL, 2019.
[3] Alakuijala, Minttu, et al. Learning reward functions for robotic manipulation by observing humans. ICRA 2023


### In the remaining global response, we would like to continue the descriptions of Thawing and CarWash environments, asked by R2 FZmh.

> Environment specifications (state and action space, their meanings, and horizon)

- Thawing:
  - state space (12d), consisting of agent position (2d), agent direction (1d), fish position (2d), fish frozen state (1d), sink position (2d), refrigerator position (2d), refrigerator state (1d), timestamp (1d)
  - action space (7d), consisting of
    - move towards 1) the refrigerator, 2) the sink, 3) the fish,
    - 4) open, or 5) close the refrigerator,
    - 6) pickup, or 7) drop the fish
  - given horizon: 100
  - minimal # of time steps required to finish the task: 11
- CarWash:
  - state space (18d), consisting of agent position (2d), agent direction (1d), rag position (2d), rag state (2d), sink position (2d), sink state (1d), bucket position (2d), soap state (2d), car position (2d), car state (1d), timestamp (1d).
  - action space (10d), consisting of
    - move towards 1) the rag, 2) the sink, 3) the soap, 4) the bucket, 5) the car
    - pick up 6) the rag, 7) the soap
    - drop 8) the rag, 9) the soap
    - toggle the sink 10)
  - given horizon: 300
  - minimal # of time steps required to finish the task: 31

---

### Decision · Program_Chairs · 2023-09-21

**Decision:**

Accept (poster)

**Comment:**

The reviewers have laregly formed a positive consensus, while suggesting a number of suggestions and in particular requested further comparison and experimentation. This appears to be an already high-quality paper that will be improved by incorporation of the author's rebuttal replies and additional results (1-pager). I recommend acceptance, while requesting the authors take extra care to fully integrate the updated(improved) story into all aspects of the final paper.